ecology, computational biology, theoretical biology

invasion, cross-feeding, invasional meltdown, community ecology, microbial ecology, biotic resistance

**Author for correspondence:**
Cristina M. Herren
e-mail: cristina_herren@hms.harvard.edu

†Present address: 310 Countway Medical Library, 10 Shattuck St, Boston, MA 02115, USA.

One contribution to a Special Feature 'Application of ecological and evolutionary theory to microbiome community dynamics across systems' guest edited by Dr James McDonald, Dr Britt Koskella, Professor Julian Marchesi.

# Disruption of cross-feeding interactions by invading taxa can cause invasional meltdown in microbial communities

Cristina M. Herren[1,2,3,†]

[1]Harvard Data Science Initiative, Harvard University, Cambridge, MA, USA
[2]Department of Biostatistics, Harvard T. H. Chan School of Public Health, Boston, MA, USA
[3]Department of Biomedical Informatics, Harvard Medical School, Boston, MA, USA

CMH, 0000-0001-5672-7506

The strength of biotic interactions within an ecological community affects the susceptibility of the community to invasion by introduced taxa. In microbial communities, cross-feeding is a widespread type of biotic interaction that has the potential to affect community assembly and stability. Yet, there is little understanding of how the presence of cross-feeding within a community affects invasion risk. Here, I develop a metabolite-explicit model where native microbial taxa interact through both cross-feeding and competition for metabolites. I use this model to study how the strength of biotic interactions, especially cross-feeding, influence whether an introduced taxon can join the community. I found that stronger cross-feeding and competition led to much lower invasion risk, as both types of biotic interactions lead to greater metabolite scarcity for the invader. I also evaluated the impact of a successful invader on community composition and structure. The effect of invaders on the native community was greatest at intermediate levels of cross-feeding; at this 'critical' level of cross-feeding, successful invaders generally cause decreased diversity, decreased productivity, greater metabolite availability, and decreased quantities of metabolites exchanged among taxa. Furthermore, these changes resulting from a successful primary invader made communities further susceptible to future invaders. The increase in invasion risk was greatest when the network of metabolite exchange between taxa was minimally redundant. Thus, this model demonstrates a case of invasional meltdown that is mediated by initial invaders disrupting the metabolite exchange networks of the native community.

## 1. Introduction

Cross-feeding, wherein one individual consumes a metabolic product of a different individual, is ubiquitous in microbial communities [1]. Stable cross-feeding relationships evolve spontaneously even when a single strain of bacteria is grown in the laboratory; in a well-studied example where a single genotype of *Escherichia coli* is grown in glucose, a second genotype capable of consuming acetate, a waste product, eventually evolves and coexists alongside the original genotype [2]. In this case, a mutation allowing an *E. coli* cell to consume the unexploited acetate resource confers a fitness advantage. The evolution of novel bacterial genotypes capable of cross-feeding has been observed and reproduced under a variety of laboratory conditions [3–5], demonstrating the widespread prevalence of cross-feeding even in simple microbial communities. However, cross-feeding is not well studied in the context of theoretical community assembly models, perhaps because many of these models were developed with macro-ecological systems in mind, where cross-feeding is comparatively rare.

The simple example of the spontaneous evolution of cross-feeding in a culture of *E. coli* demonstrates how cross-feeding can alter community structure. The number of functionally distinct taxa in this case increases from one to two, and the total density of cells may increase as the acetate-consuming genotype is able to subsist on a resource that would otherwise not be consumed. Thus, diversity, productivity (cell density), and metabolite concentrations would all be affected by the establishment of this cross-feeding relationship. Empirical studies have found that cross-feeding is a vital process in determining what populations can persist within microbial communities [6]. A single bacterial strain can produce dozens of metabolic by-products capable of sustaining other strains [7]. Therefore, in more complex communities, there is vast potential for cross-feeding between bacteria [8]; the number of possible cross-feeding relationships increases with the number of taxa present in the community and the number of nutrients provided in the environment. Thus, cross-feeding has the potential to alter community structure across a broad range of microbial ecosystems, and these structural changes may have cascading effects on community stability and function.

Incorporating cross-feeding into mathematical models can be computationally challenging, which may account for why much of the theoretical development of this topic has been recent. Incorporating cross-feeding into models introduces many additional parameters, as these models must track the concentrations of each metabolite in the environment and within cells, in addition to the exchanges of each metabolite between cells. Previous theoretical models studying the effects of cross-feeding on community assembly have largely focused on whether communities with cross-feeding are stable and how these relationships affect the diversity of communities (e.g. [6,9,10]). For example, in classical ecological models, there is a paradigm that only one consumer can persist for each resource present in an ecological community [11]. However, recent theoretical models have found that cross-feeding can dramatically increase the diversity of taxa, even in a homogeneous environment [6,10,12]. Furthermore, multiple different types of models have found that introducing cross-feeding into communities can result in a new stable community composition [13–15]. However, fewer studies have examined how the strength of cross-feeding relationships alters other emergent properties, such as susceptibility to invasion.

The model presented here uses a metabolite-explicit mathematical simulation to study how cross-feeding between microbial taxa affects the ability of an introduced taxon to invade the community. First, I study how cross-feeding alters the assembly of microbial communities containing randomly generated taxa. Then, I investigate how cross-feeding networks and community structure mediate the ability of an invader to join an established community. In the case of a successful invader, I evaluate how the introduced taxon alters the composition and cross-feeding network of the microbial community. Finally, I ask whether a successful primary invader can lead to 'invasional meltdown' by making the community more susceptible to future invaders [16]. Thus, this modelling approach studies the interplay between community structure, biotic interactions, and invasion history in determining the susceptibility of a microbial community to invasion.

## 2. Material and methods

I constructed a mathematical model consisting of resident taxa, invading taxa, and the metabolites required for cell reproduction. A general mathematical formulation of the model is available in the electronic supplementary materials, and is described here. Taxa interact through competition for metabolites in the environment and through cross-feeding, defined here as the directed transfer of metabolites between taxa. Of all possible metabolites in the model ($m$), each taxon requires a randomly assigned unique subset of $n$ metabolites for growth, giving each a distinct ecological niche. At the beginning of each model run, $x$ native taxa were introduced into the community. For example, for the models presented here, there were 20 native taxa, each with an abundance of 50, at the start of the simulation. There were eight possible metabolites, and each taxon required five of those eight metabolites for reproduction. Thus, there were a total of eight choose five (equal to 56) distinct niches that taxa could occupy. From these 56 niches, 20 niches were randomly assigned to the native taxa, and one was assigned to the invasive taxon; this yielded 56 choose 20 (upwards of 100 trillion) combinations of possible metabolite requirements for the native taxa. Each taxon also excretes a subset of $q$ metabolites, which do not overlap with its $n$ required metabolites. The 'input' metabolites are a set of $n$ metabolites that entered the environment at the beginning of each timestep, and one of the $x$ native taxa had metabolite requirements that matched the input metabolites.

Cross-feeding in the model was implemented as one taxon directly transferring its excreted metabolites to another taxon that required those metabolites. All possible unidirectional metabolite transfers were identified by looking at which metabolites were excreted and required by all taxa; a random fraction (given by the cross-feeding parameter $p$) of these possible metabolite transfers were implemented as cross-feeding relationships in the model. The cross-feeding step occurred separately from competitive uptake of metabolites from the environment. Other parameters in the simulation model include the average competition coefficient ($c$), variability in competition coefficients among taxa ($v$), an input rate for metabolites ($i$), and a flushing rate for metabolites and cells ($f$). Competition coefficients for native taxa were drawn from a normal distribution with mean $c$ and a standard deviation of $v$. The initial abundance of all native taxa when initializing the model was 50, and this was also the abundance at which the invader was introduced.

Each timestep begins with input metabolites entering the environmental pool (figure 1*b*). Taxa then compete for these metabolites, with uptake rates governed by their competition coefficients, which quantify scavenging efficiency. Each individual is able to store 1 unit of each required metabolite, and reproduction occurs when the individual procures 1 unit of all its required metabolites. Metabolite uptake from the environment is allocated proportionally among taxa in accordance with each taxon's demand for the metabolite; demand for a metabolite is calculated as the number of individuals needing the metabolite multiplied by their respective competition coefficients. If there is metabolite scarcity (meaning that total demand for metabolites exceeds availability of metabolites), metabolites are allocated among taxa in proportion to the demand of each taxon. I assume for simplicity that metabolite uptake among individuals in a population is arranged to maximize biomass production [17]. Population growth is limited by whatever metabolite is most scarce in the population. The reproducing individuals (those having acquired all necessary metabolites) also excrete one unit of each of the metabolites in their excretion profile. If these individuals are from taxa participating in cross-feeding, the excreted metabolites are preferentially available to the recipient taxon; in this case, the metabolites are directly transferred to the recipient without being available for competitive uptake. If the reproducing

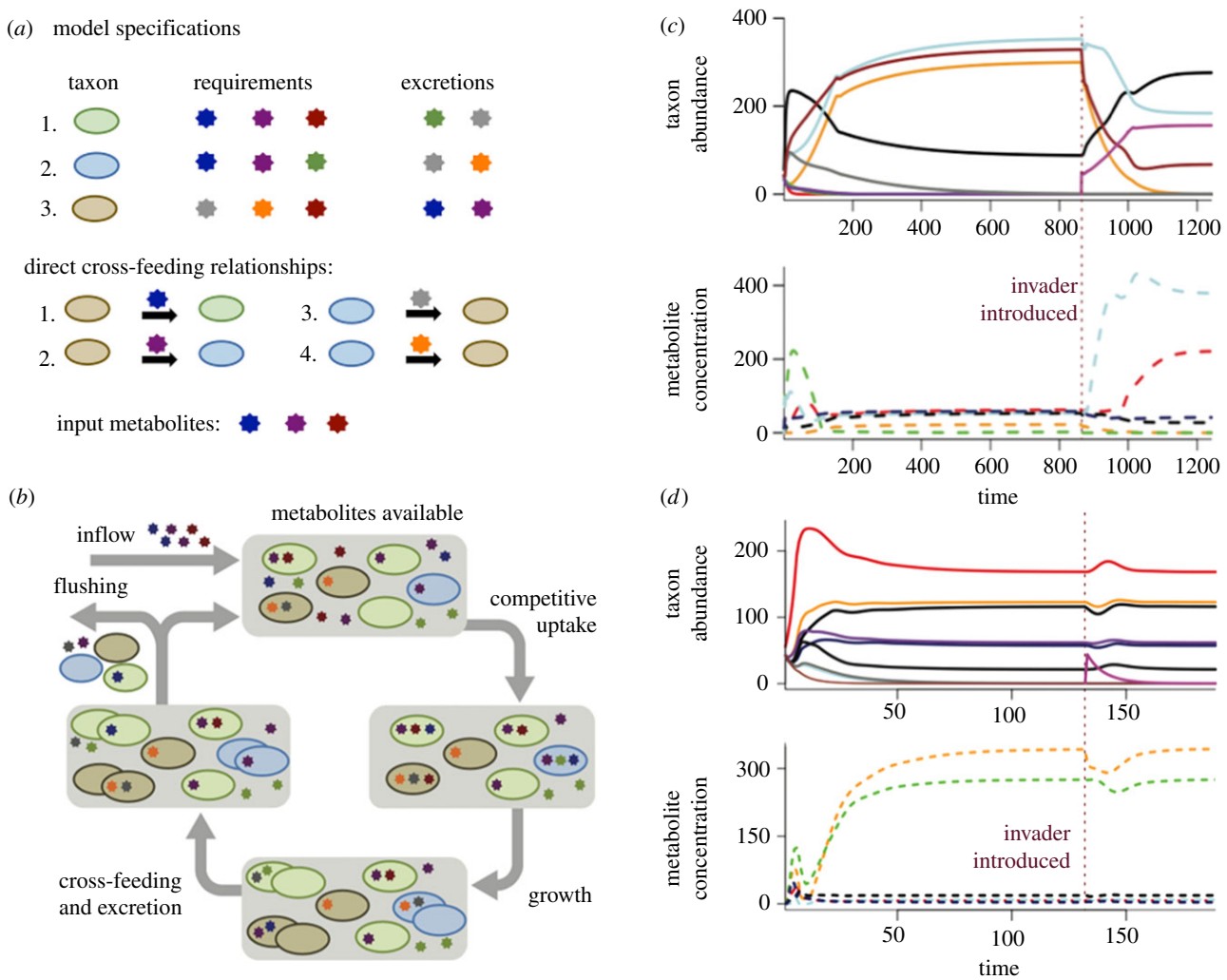

**Figure 1.** Design and output of simulation model studying invaders in microbial communities. Panel (*a*) gives model specifications for a simplified version of the cross-feeding model, containing three taxa, which is depicted in panel (*b*). Panel (*b*) shows the processes that occur during each timestep of the model. Different metabolites are represented by differently coloured stars. Different taxa are represented by differently coloured ovals. When a cell acquires one unit of each of its required metabolites, it reproduces and also excretes its given metabolites. In this example, the native community of three taxa has reached equilibrium. Panels (*c*,*d*) show results of model simulations, tracking both taxon abundances and the concentration of each metabolite in the environment through time. Panel (*c*) shows a successful invasion, where the invading taxon (pink line) persists in the community, whereas in (*d*), the invader is excluded from the community. Red dashed lines indicate the time point when the invader is introduced. (Online version in colour.)

taxon has more than one cross-feeder, the excreted metabolites are divided equally among the recipient taxa. Any excreted metabolites that are not part of cross-feeding relationships enter the environmental pools of metabolites. Thus, this model also allows for 'indirect' cross-feeding, wherein taxa can consume metabolites from the environment that were produced by a different taxon. However, the term 'cross-feeding' in this paper refers to direct metabolite transfers between taxa. Finally, a proportion $f$ of individuals and environmental metabolites are flushed from the system.

The invader was introduced after the community of resident taxa equilibrated (figure 1*c*,*d*). Equilibrium was determined as when the maximum change in any taxon's population was less than 0.001 between timesteps. The invader had a fixed competition value of 0.9 in all simulations (generally larger than that of native taxa), and did not have any cross-feeding relationships. The lack of cross-feeding relationships is the primary way in which the invader differs from native taxa. There are multiple reasons why invasive taxa were not allowed to cross-feed in the model. First, I reasoned that cross-feeding relationships often need time to develop (e.g. time for proper spatial configuration [18], construction of nanotubes [19], or within-host coevolution [20]), and that an invading taxon would therefore

have no pre-existing methods of directly acquiring metabolites. Additionally, many studies of invasive taxa have concluded that invasive taxa differ from native taxa in their biotic interactions (as reviewed in [21]). The lack of cross-feeding relationships for invaders differentiates the biotic interactions of invaders from those of native taxa. Finally, the invader was given a relatively high competition coefficient because strong competitive ability can be another characteristic trait of invasive taxa [22].

After the invader was added, the simulation continued until the community again reached equilibrium. A successful invader changes the abundances of native taxa by introducing additional competition for metabolites. After the model equilibrates, a second invader with a different, randomly chosen metabolite profile was added, and again the model was run until equilibrium. At each of these three equilibria (without invader, after the first invader, and after the second invader), I recorded properties of the community and properties of the cross-feeding network established between community members (figure 1*b*). The community-level outcomes recorded were persistence of the invaders, total individuals in the community, number of taxa present in the community, and the number of metabolites in the environment at equilibrium (table 1). Successful invaders

**Table 1.** Input parameters and measured outputs for the cross-feeding model.

| input parameters | value |
| --- | --- |
| maximum number of taxa in community ($x$) | 20 |
| number of possible metabolites ($m$) | 8 |
| number of metabolites required by each taxon ($n$) | 5 |
| number of metabolites excreted by each taxon ($q$) | 3 |
| flushing rate of cells and metabolites ($f$) | 0.1 |
| metabolite input rate ($i$) | 200 per timestep for each metabolite |
| proportion of direct cross-feeding relationships ($p$) | 0.0–0.5 in increments of 0.01 |
| mean competition coefficient of native taxa ($c$) | 0.5–0.8 in increments of 0.01 |
| standard deviation of competition coefficients ($v$) | 0.3 * mean competition coefficient ($c$) |
| **measured model outputs** | **definition** |
| persistence of invader | an invasion was deemed successful if the invader had an abundance greater than 1 at model equilibrium |
| total individuals | sum of all individuals from all taxa at equilibrium |
| taxa coexisting | number of taxa with at least 1 individual present at equilibrium |
| metabolites at equilibrium | sum of all metabolites present in the environment at equilibrium |
| metabolites traded | sum of all metabolites directly exchanged through cross-feeding |
| flows per taxon | number of direct cross-feeding relationships per taxon |
| redundancy of limiting flows | average number of cross-feeding relationships that provide the growth-limiting nutrient to each taxon |

were counted in the total number of individuals and total number of taxa. The network-level outcomes recorded were the number of metabolites traded during each timestep, the average number of cross-feeding relationships (abbreviated in figures as 'flows' of metabolites) for each taxon, and the average number of cross-feeding relationships (again, metabolite 'flows') providing each taxon's growth-limiting nutrient (table 1). Finally, I also tested whether the second invader could persist in the absence of the first invader by resetting the community to its first equilibrium and adding only the second invader. I evaluated these model outputs while changing the proportion of cross-feeding relationships and the degree of competition present between taxa.

Parameter values used in model simulations can be found in table 1. Any randomly generated competition values below 0.1 were set to 0.1, as to minimize the outcome that no taxa were able to persist in the community. Results were qualitatively similar regardless of the number of taxa used in the simulation ($x$), so long as there were sufficiently many taxa (at least 8–10). I generated 5000 simulated communities for each combination of competition coefficient and cross-feeding proportion, resulting in a total of 7 905 000 simulated communities. Thus, there are 5000 values of each model output for each set of parameters evaluated. In a small fraction of runs (3.5%, on average), the model resulted in a stable limit cycle or did not equilibrate within 40 000 timesteps, and these runs were discarded.

## 3. Results

The proportion of possible cross-feeding relationships present in a community strongly influenced community structure and connectivity of the metabolite exchange network. Higher prevalence of cross-feeding was related to increased diversity,

increased productivity (more individuals in the community), fewer metabolites in the environment, and increased metabolite exchange between individuals.

Biotic interactions between taxa within the native microbial community were strong determinants of whether an invading taxon could persist in the community. Invasive taxa were most successful when both competition and cross-feeding within the resident community were weak (figure 2$a$). When competition and cross-feeding were at their lowest values, invaders were successful in nearly every community, whereas invaders succeeded less than 1% of the time in communities with the maximum competition and cross-feeding values. Secondary invaders (those introduced at model equilibrium after the first invader) were more successful than primary invaders across all values of competition and cross-feeding (figure 2$b$). The largest discrepancy between the success of secondary invaders versus the success of primary invaders was at intermediate cross-feeding values (figure 2$c$).

I then evaluated whether communities that were susceptible to one invader were more susceptible to other invaders (figure 2$d$–$f$). I tested the ability of the same independent invader (meaning, with the same metabolic profile) to join a community either in the absence of a primary invader (figure 2$d$) or after a primary invader had previously established in the community (figure 2$e$). In both cases, communities that could be invaded by one type of invader were also more susceptible to a different type of invader. However, the presence of a primary invader within a community increases susceptibility to a secondary invasion, and this invasional meltdown was most likely to occur at intermediate levels of cross-feeding (figure 2$f$). Additionally, this analysis demonstrates that the metabolite profile of the invader is a

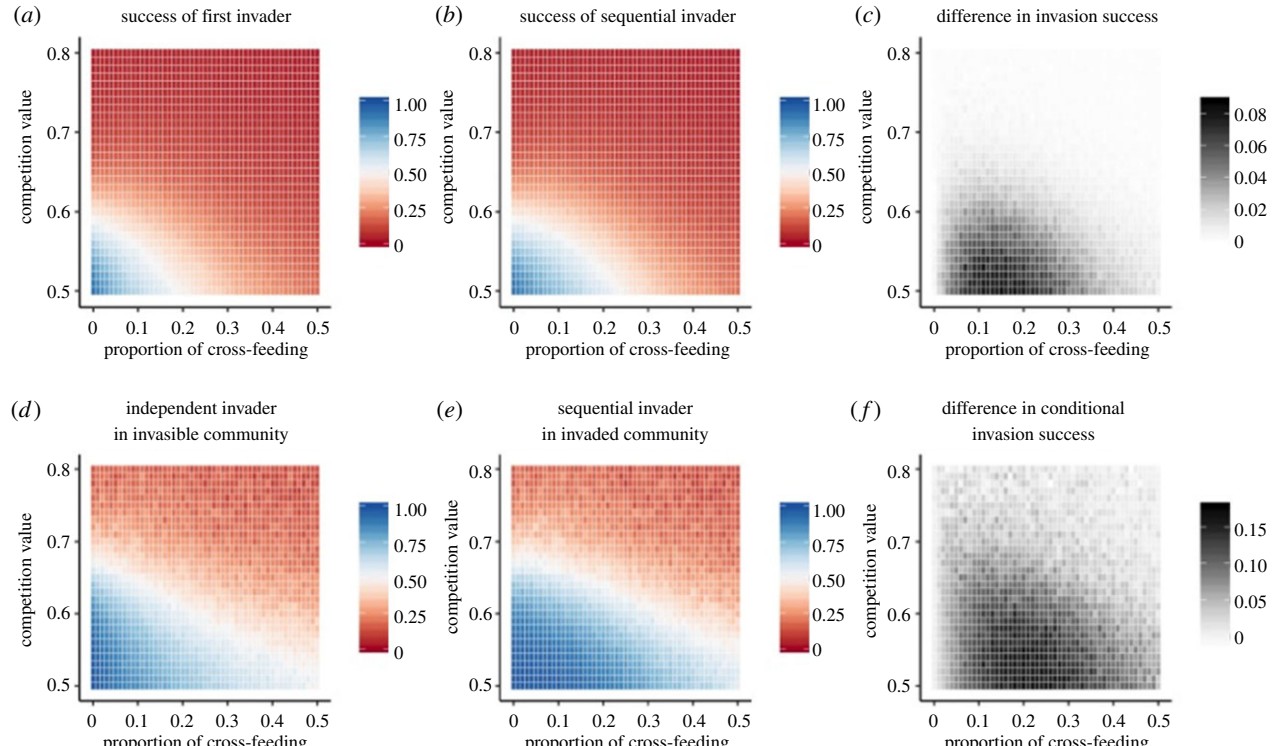

**Figure 2.** Primary and secondary invasion success across all communities (*a–c*) and within communities susceptible to invasion (*d–f*). Primary invaders (i.e. the first invader introduced) are highly successful when cross-feeding and competition are low, but quickly become less successful as the strength of either of these interactions increases (*a*). A sequential invader (introduced after the primary invader) is more successful than the primary invader (*b*). The difference in invasion success (success of sequential invader minus success of primary invader) is greatest at intermediate levels of cross-feeding and low levels of competition (*c*). I isolated communities susceptible to the primary invader, and tested whether a different invader would be able to succeed there (*d*). Communities that were invasible by one invader were generally much more susceptible to a different invader. However, those same communities were more susceptible to a sequential invader, in comparison to a primary invader (*e*). Again, the difference in invasion success as a result of the presence of the first invader was greatest at intermediate levels of cross-feeding (*f*). (Online version in colour.)

determinant of invasion success, as communities that are invasible by one invader are not completely susceptible to a different invader.

After observing that the invasional susceptibility increased most at intermediate degrees of cross-feeding, I looked for a mechanism that might cause this pattern. I compared the difference in susceptibility with the average redundancy in the cross-feeding relationships providing each taxon's limiting resource (figure 3*a*). The limiting nutrient is an important quantity to track in this model, because reducing the supply of the limiting nutrient hinders population growth, whereas this is not necessarily true for non-limiting metabolites. As the redundancy of the limiting resource flows increases, it is less likely removing a single taxon will lead to an absence of a limiting nutrient for another taxon (figure 3*a,b*). I found that increased susceptibility to invasion was most common when the average number of taxa providing each limiting resource was less than 1 (figure 3*a*). Thus, communities with high redundancy in metabolite exchanges were strongly protected from both primary and secondary invaders.

To further investigate why invasional meltdown was the strongest at intermediate levels of cross-feeding, I compared the community and network structures of uninvasible, invasible, and invaded communities across different values of cross-feeding. If any of these community or network properties contributed to increased susceptibility to invasion, the property should be different between uninvasible and invasible communities. Furthermore, properties affecting invasion

risk should also be impacted by the presence of a successful invader (because these communities were shown to be more invasible). I found this pattern to some degree in all six of the simulation properties studied (figure 4).

There were consistent differences in community properties between uninvasible and invasible communities (figure 4*a–c*), although metabolite network properties showed more pronounced differences between uninvasible and invasible communities (figure 4*d–f*). Uninvasible communities were generally more diverse than invasible communities. An invading taxon could reduce overall diversity if the invader caused the loss of more than one taxon from the original community (figure 4*a*). However, at high levels of cross-feeding, a successful invader generally did not displace any taxa. Invasible and uninvasible communities did not consistently differ in their total number of individuals, though invasible communities at moderate levels of cross-feeding generally had fewer individuals (figure 4*b*). However, the number of metabolites present at equilibrium was consistently different between uninvasible, invasible, and invaded communities; uninvasible communities had relative metabolite scarcity, and invaded communities had comparatively high metabolite availability (figure 4*c*). Additionally, uninvasible communities had the largest amount of metabolites exchanged through cross-feeding, whereas invaded communities had the lowest amount of exchanged metabolites (figure 4*d*). Two other measures of the strength of the cross-feeding network, the number of metabolite flows per taxon (figure 4*e*) and the number of flows providing the limiting metabolite (figure 4*f*) also

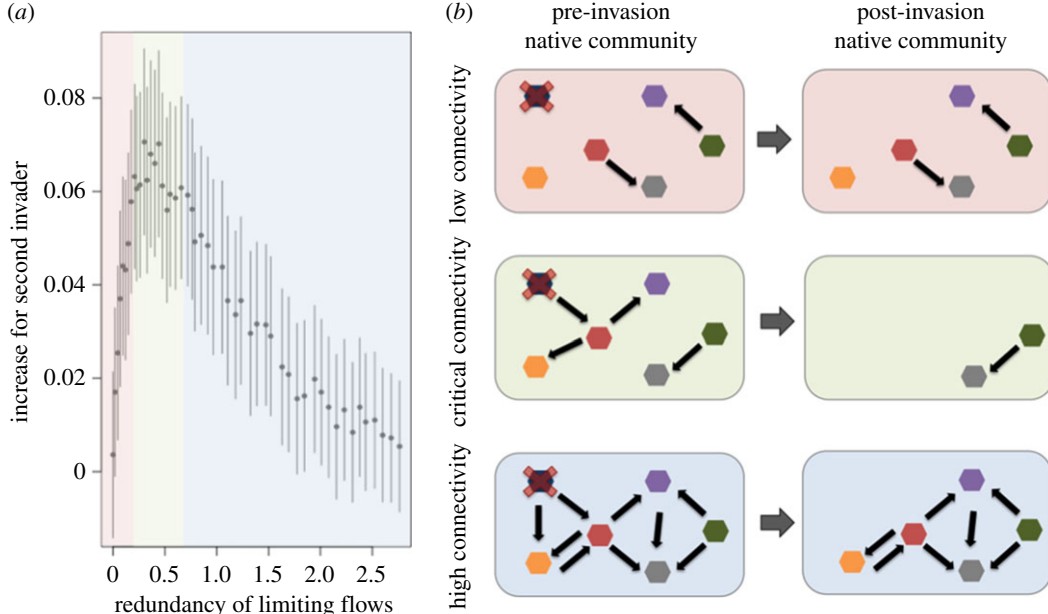

**Figure 3.** The difference in success between a first invader and a secondary invader is greatest when there are intermediate amounts of metabolite exchanges (*a*). The redundancy of limiting flows measures the average number of cross-feeding relationships that provide each taxon with its growth-limiting metabolite. Bars show 95% confidence intervals for the difference in invasion success between a first and second invader, calculated using the standard error for binomially distributed variables. Shaded areas approximate regions of low cross-feeding connectivity (red), intermediate or 'critical' connectivity where the invasion difference is highest (green), high connectivity (blue). Panel (*b*) gives an example of how invaders have maximal impact at critical connectivity. This illustration shows the same community with different cross-feeding dependencies. If a single native taxon is competitively excluded after invasion (blue taxon with red X), it is likely that no other taxa are singularly dependent on this taxon at low or high connectivity (red and blue shaded communities, respectively). However, at critical connectivity, removing a single taxon causes all taxa with downstream dependencies to also become extinct (green communities). The data in panel (*a*) are from simulations where the mean competitive values of the native communities are fixed at 0.55. (Online version in colour.)

showed that a successful invader weakened cross-feeding networks. Similarly, communities that assembled with higher redundancy in metabolite exchanges were less invasible (figure 4*e,f*).

Finally, I evaluated how strongly an invader altered a community as a function of the degree of competition and cross-feeding present in the native community. Across all metrics studied, I found that an invader had the strongest effect on community structure (figure 5*a*–*c*) and networks of metabolite exchange (figure 5*d*–*e*) at intermediate levels of cross-feeding. However, each of the six metrics had subtle differences in how invasion impacted them at different competition and cross-feeding values. For changes in community diversity, taxa were generally excluded at intermediate levels of cross-feeding, but added at higher levels of cross-feeding; this was consistent across all competition values (figure 5*a*). It was possible for an invader to lead to increased diversity by excreting novel metabolites into the environment, thereby creating new niches for taxa to occupy. In this case, native taxa that were previously counted as absent (having a population of less than 1) increased in abundance to join the community. Similarly, most communities showed declines in the density of individuals after invasion (figure 5*b*), but these losses of individuals were more extreme at low competition values. Conversely, at high competition and cross-feeding values, there was generally a gain in the total number of individuals after invasion. The total number of metabolites in the environment at equilibrium increased after invasion at high levels of competition (figure 5*c*). However, metabolite concentrations generally decreased at high and low levels of cross-feeding, especially when competition values were also low.

The networks of metabolite exchanges between taxa were overwhelmingly weakened by the introduction of an invader (figure 5*d*–*e*). All of the network properties showed decreases in connectivity/redundancy at intermediate levels of cross-feeding, regardless of the strength of competition. Additionally, the small impact on cross-feeding networks at low values of cross-feeding stemmed primarily from the fact that there was a minimal established network in this parameter range, and thus the maximum possible disruption to the network was small. However, there were fine-scale differences in how invaders affected these three aspects of cross-feeding networks. The number of metabolites traded (figure 5*d*) was negatively affected at the lowest threshold of cross-feeding, but was minimally affected at very high levels of cross-feeding. Furthermore, the number of metabolite flows providing limiting nutrients was affected at a lower threshold of cross-feeding than the total number of metabolite flows, confirming that quantities of limiting and non-limiting metabolites affected taxa differently.

## 4. Discussion

These studies of invasion within simulated microbial communities show that cross-feeding is a strong determinant of microbial community assembly and of the potential for new taxa to enter the community. Stronger biotic interactions between resident taxa, whether from cross-feeding or competition, resulted in lower rates of invasion (figure 2). After accounting for the effects of interactions within communities, invasion was more likely when metabolites were abundant and diversity was low (figure 4). However, network properties

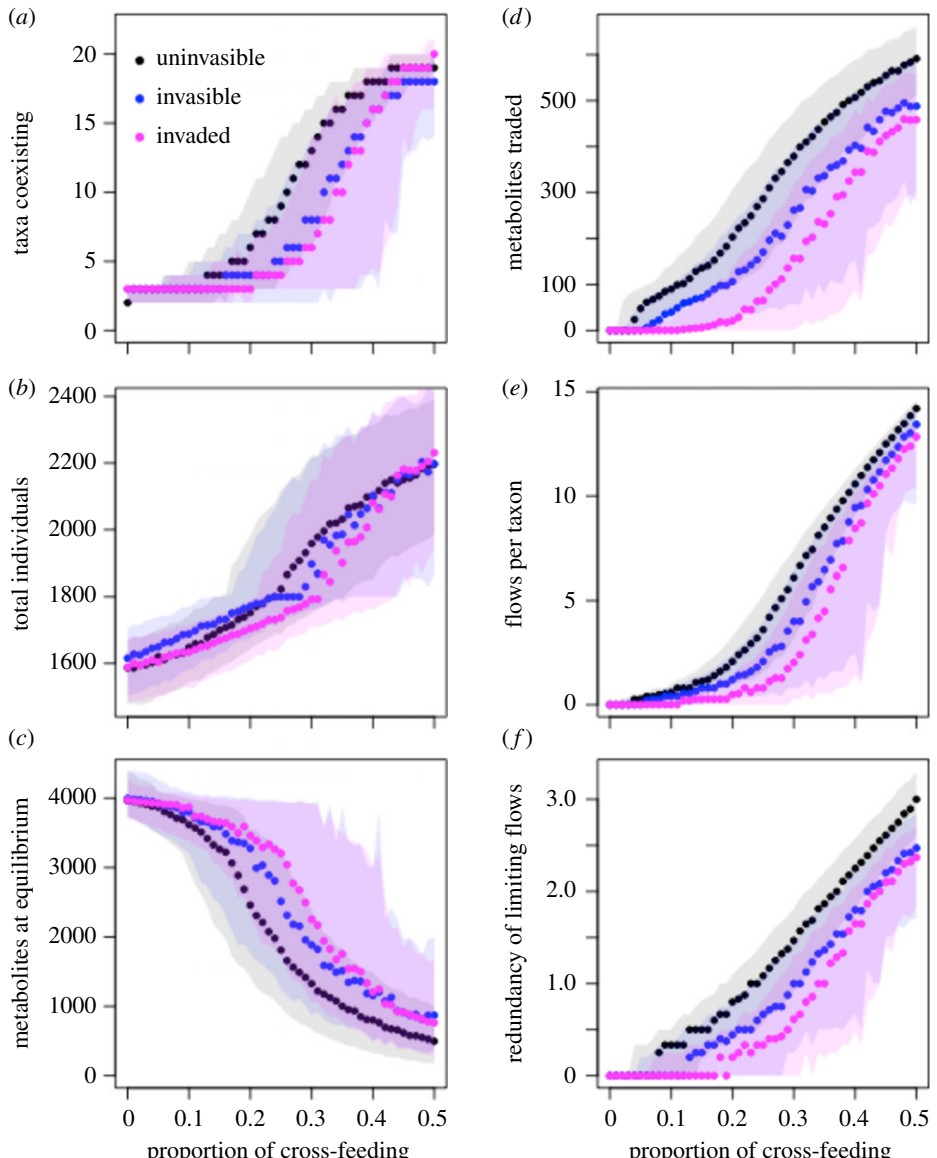

**Figure 4.** Properties of uninvasible (black), invasible (blue), and invaded (pink) communities assembled under differing levels of cross-feeding. Each panel shows one aspect of the community structure (a–c) or metabolite exchange network (d–f). Solid points with shaded envelopes show the median values and interquartile range. Properties that contribute to invasional meltdown would be expected to be different between uninvasible and invasible communities, and should be further shifted after a successful invasion. All communities shown have the same mean competition coefficient ($c = 0.55$). (Online version in colour.)

were more reliable indicators of invasibility than community structure; there was strong differentiation between invasible and uninvasible communities based on the number of metabolites exchanged and the redundancy of flows providing limiting nutrients, with invasible communities having weaker cross-feeding networks (figure 4). Invading taxa had the greatest impact on the resident communities at intermediate levels of cross-feeding and competition (figure 5). In this case, invasion was somewhat common (approx. 20–50% success rate, figure 2), and caused declines in diversity and productivity of the community, leading to more unused metabolites. Additionally, all aspects of the cross-feeding network were weakened. However, it was possible for invaders to increase overall diversity, and this result was most common at the highest levels of cross-feeding (figure 5).

This study demonstrates that invasional meltdown can occur as a result of initial invaders disrupting the cross-feeding network of a native set of taxa, thereby making the community more susceptible to another invader. Invasional meltdown, defined here as an increased success rate of a

secondary invader, was observed across all parameter space evaluated, but was the strongest at intermediate levels of cross-feeding (figure 3). Thus, there was a critical level of cross-feeding at which communities were most prone to undergo dramatic shifts, if disturbed (figure 5). Critical connectivity occurs when there are many taxon contingencies but minimal redundancy, such that a disruption in the network has a cascading effect of removing taxa/individuals (figure 3). When an invader is added to a community, it can directly exclude individuals by increasing competition for a metabolite to sufficiently high levels that the resident taxa cannot persist. If a competitively excluded taxon provided limiting metabolites to other taxa, those taxa could be secondarily excluded as a result of the loss of their requisite cross-feeding relationships (figure 3). This community collapse does not occur at sufficiently low or high levels of network connectivity. At low levels of cross-feeding, the pre-existing cross-feeding network is minimal, so there is a low probability of an invader disrupting a chain of cross-feeding relationships (figure 3b). Conversely, at high

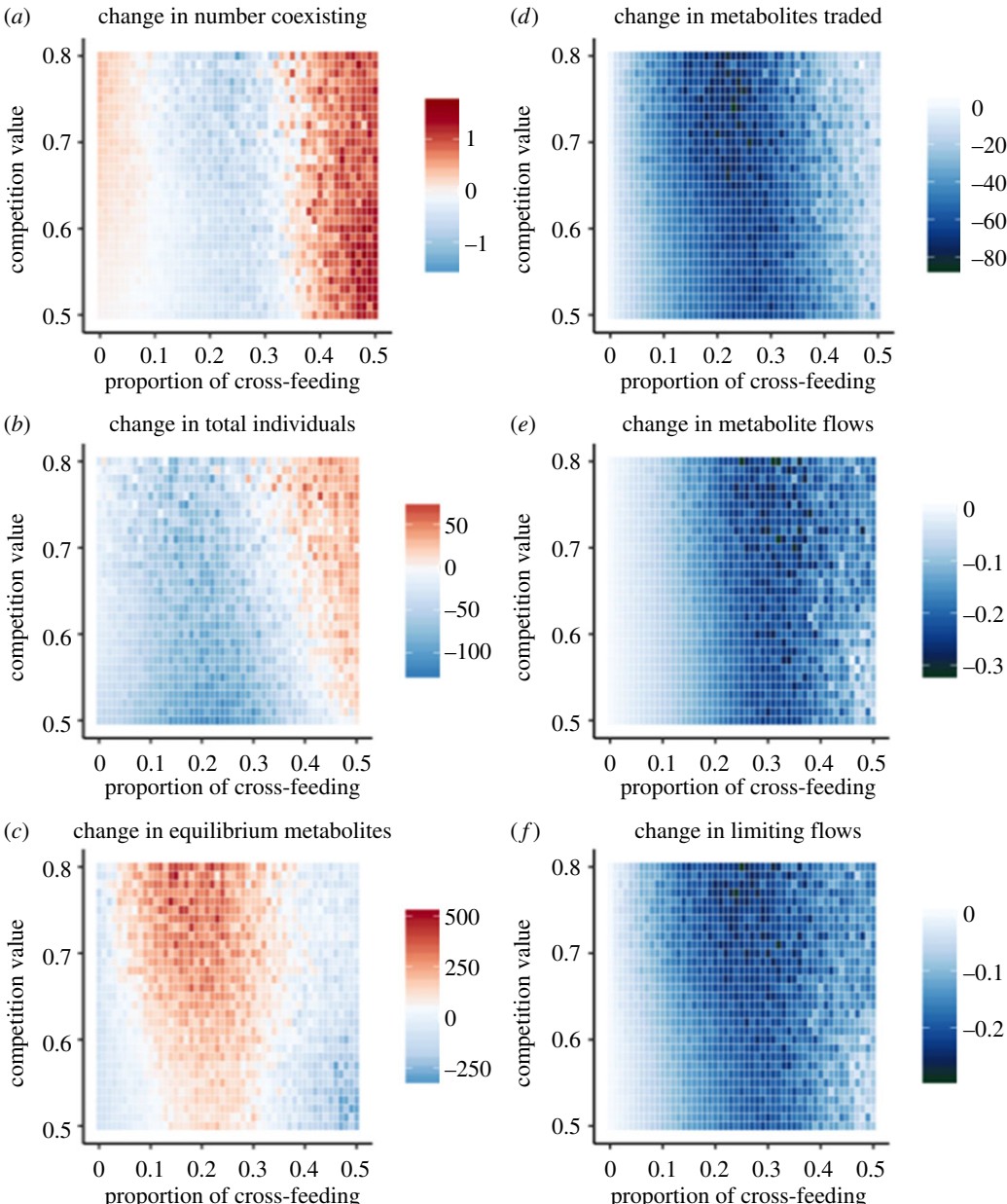

**Figure 5.** Changes in community and network properties as a result of a successful invasion. Each panel shows the median change in community (*a–c*) or metabolite exchange network (*d–f*) properties when comparing the pre-invasion community to the post-invasion community. Warm colours indicate increases, whereas cool colours indicate decreases. Across all properties considered, there were strong changes at intermediate levels of cross-feeding. (Online version in colour.)

metabolic connectivity, the cross-feeding network is redundant, so multiple taxa provide the same function; thus, high levels of cross-feeding protect against the domino effect of species loss (figure 3*b*).

Invasive taxa often differ from native taxa in their interactions with other organisms [23]. In many cases, these altered biotic interactions contribute to the success of the invader [24,25]. The assumption in this model that invaders cannot cross-feed is the primary way in which the invasive taxa are differentiated from native taxa. Although the invaders' competition coefficients were relatively high, they were still within the range of values that could be assigned to native taxa. This lack of cross-feeding by the invader proved crucial to the phenomenon of invasional meltdown; when allowing the invader to have the same cross-feeding dynamics as the native taxa, there was no increased susceptibility to future invasion after a primary invasion (electronic supplementary material, figure S1). Furthermore, a successful invasion under these circumstances was less disruptive to

overall community structure (electronic supplementary material, figures S2 and S3). Thus, these sensitivity analyses show that even a single taxon that does not participate in cross-feeding strongly affects the entire microbial community. However, the model was much less sensitive to assumptions about how cross-feeding was implemented among native taxa, as results were qualitatively similar when native taxa were allowed to be differentially good or poor at obtaining metabolites through cross-feeding (electronic supplementary material, figures S4, S5, S6). Thus, the conclusions from this study apply primarily to cases where the invader is not well integrated into metabolite exchanges among the native community. Future models might use different criteria to differentiate an invader from a native taxon, such as specifying unique metabolite requirements for the invader, or introducing distinctions between native and invasive taxa in their indirect cross-feeding.

This study adds to the long history of theoretical literature investigating how the strength of biotic interactions affects

community structure and stability. The idea that strong interspecies interactions within native communities can mediate susceptibility to invading taxa has become known as 'biotic resistance' [26]. Biotic resistance can occur through many mechanisms (reviewed in [23]), including eliminating open niches through strong competition between resident taxa [24]. Additionally, the strength of biotic interactions between the native community and an invading taxon can determine the outcome of an invasion [25]. However, few studies have investigated cross-feeding as a mechanism of biotic resistance, perhaps because cross-feeding is relatively uncommon outside of microbial communities. In this study, the strength of biotic interactions was related to both invasion risk and the magnitude of the effects of a successful invader. However, the probability of invasion (figure 2) and the consequences of invasion (figure 5) were decoupled, in the sense that they were maximized at different strengths of biotic interactions.

The organization of microbial communities differs from the organization of macro-scale communities [27], and the prevalence of cross-feeding in microbial communities may be one reason why these communities are structurally distinct. First, cross-feeding is one possible contributing factor to the high diversity of microbial communities, frequently referred to as the 'paradox of the plankton' [28]. The paradox arises because more taxa coexist than there are nutrients in the system, which violates the competitive exclusion principle [21] and the rule that only one taxon should persist per possible limiting resource [11]. One feature that distinguishes these results from some previously published cross-feeding studies is that the number of taxa in this study can exceed the number of metabolites. In the simulations here, multiple taxa can procure the same limiting metabolite from different sources, which gives an example of how competitive exclusion may be avoided. Additionally, cross-feeding has been shown to enable the stable coexistence of microbial communities grown in the laboratory. In the Long-term Evolution Experiment, multiple genotypes of *E. coli* have coexisted in a homogeneous culture [29], with evidence of cross-feeding between some genotypes [5]. Cross-feeding had previously been proposed as a mechanism for the repeated co-occurrence of taxa across varied environments [30], and these models agree with this possibility. Similarly, a recent study of naturally occurring marine microbial communities showed that collections of taxa synchronously rose and fell in abundance at the daily time scale, with biotic interactions between taxa as one proposed mechanism of the cohesive dynamics of these subcommunities [31]. Thus, there is

growing evidence that microbial communities contain modules of taxa with linked abundance patterns, and the basis for these subcommunities is metabolic contingency via cross-feeding.

In addition to linking cross-feeding to changes in community structure, this study further found that cross-feeding can alter community function. Previous empirical studies have similarly suggested that cross-feeding dependencies shape emergent functions of communities. For example, cross-feeding can lead to succession of taxa within a microbial community [32], thereby altering the metabolic capacity of a community and the potential for degrading compounds in the environment [33]. Another way that biotic interactions might impact community functionality can be observed when two microbial communities intermix. Instead of resembling a proportional mixture of the starting communities, the community that emerges after mixing often resembles one of the initial communities more strongly; the correlated successes of taxa at the community scale has been termed 'community coalescence' [34]. Furthermore, the initial community that is dominant in the resulting mixture often has disproportionate contribution to overall community function and metabolism [35]. One proposed mechanism for this cohesiveness is that established interactions, such as cross-feeding, reinforce community structure [36]. In this case, biotic interactions link the success of co-dependent taxa, and these modules/subcommunities of taxa collectively displace one another. This hypothesis about the cohesive force of cross-feeding also agrees with the observation that communities comprised of highly interconnected taxa have greater compositional stability [37]. In this framework, the presence of cross-feeding would both cause many taxa to show similar abundance patterns through time and would buffer against compositional change within the community. In a review of the prevalence and characteristics of microbial invasions, Litchman [22] proposed that low metabolic diversity or poor resource use efficiency may increase the susceptibility of a community to invasion. This work suggests that cross-feeding underlies these characteristics of resource use and niche availability to shape emergent community functions.

Data accessibility. This article has no additional data.
Competing interests. I declare I have no competing interests.
Funding. This work was funded by a fellowship from the Harvard Data Science Initiative.
Acknowledgements. I received helpful comments from Michael Baym, Anurag Limdi, and the Baym Lab at Harvard Medical School.

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
