## [Reviewer comments · Proceedings of the Royal Society B: Biological Sciences]

Review History

RSPB-2019-1494.R0 (Original submission)

Review form: Reviewer 1

Recommendation

Major revision is needed (please make suggestions in comments)

Scientific importance: Is the manuscript an original and important contribution to its field?

Acceptable

General interest: Is the paper of sufficient general interest?

Good

Quality of the paper: Is the overall quality of the paper suitable?

Marginal

Is the length of the paper justified?

Yes

Should the paper be seen by a specialist statistical reviewer?

No

Do you have any concerns about statistical analyses in this paper? If so, please specify them explicitly in your report.

No

It is a condition of publication that authors make their supporting data, code and materials available - either as supplementary material or hosted in an external repository. Please rate, if applicable, the supporting data on the following criteria.

Is it accessible?

Yes

Is it clear?

Yes

Is it adequate?

Yes

Do you have any ethical concerns with this paper?

No

Comments to the Author

The manuscript 'Disruption of cross-feeding interactions by invading taxa can cause invasional meltdown in microbial communities' uses a metabolite-explicit model to explore how the strength of cross-feeding in a microbial community affects the risk and consequences of invasions. This is an interesting topic, the manuscript is well-written and some interesting results are presented.

My main concern is that the manuscript lacks any mathematical description of the model and of model outcomes. Although an overview of all parameters is given (Fig. 1b) as well as some explanation in the main text (p. 6), without such a mathematical description (e.g. differential equations), fully understanding and reproducing the model, as well as comparing it to similar models (e.g. Kettle et al., 2018, Marsland et al., 2019), is very challenging. Especially as this is a purely theoretical paper, the theoretical framework should be presented in a clear and unambiguous way. I appreciate that R-code is accessible, but very few readers will go through almost 1000 lines of code to find relevant details. I also think that Fig. 1a, where a graphical overview of the model is given, can be improved, as it currently fails to visualize some important aspects of the model in a clear way. For instance, it is not clear from the figure what the metabolite requirements are for the different depicted taxa (why are there seemingly already some metabolites consumed before uptake takes place?), how taxa differ in competitive strength and how this affects their uptake, why some taxa grow while others do not, and what the cross-feeding interactions are. Further, I would suggest adding the symbols (as used in the text) describing the input parameters to Fig. 1b, and specifying (mathematically) how model output values are obtained (not all model outputs seem to be mentioned in Fig. 1b, e.g. redundancy of limiting flows?).

I do see the potential of this paper, and some interesting analysis and results are presented. However, I really need a better presentation of the modeling framework before I can start understanding and interpreting the results. In addition, I have the following specific comments/questions (with most of these reflecting the difficulties I had in understanding the model):

- I don't fully understand the cross-feeding procedure. A proportion p of all possible cross-feeding interactions is realized, but what exactly happens with the secreted metabolites that are not involved in a cross-feeding interaction? In the manuscript it says that these enter into the environment for competitive uptake. Can these directly be consumed by other taxa? If so, what is the effect of these cross-feeding interactions; is it giving priority to some taxa, before all other taxa

get the chance of consuming the produced metabolites? But these taxa that consume the metabolites in this second phase surely are also 'cross-feeders', as they consume metabolites produced by others? Is this in agreement with what we observe in microbial communities: are there fixed cross-feeding interactions, even in the presence of more competitive taxa that are also capable of consuming the involved metabolites? (due to some space structure?)

- Related to this, how does the other half of the invasion landscapes (Fig. 2) look, if the proportion of cross-feeding increases further up to 1? Why is 0.5 chosen as the maximum? When set at 1, this would correspond to a scenario where all taxa can directly consume the secreted metabolites? Would this decrease the number of coexisting taxa (as only the competitive taxa can persist), and increase the invasion success (as the invader is even more competitive, and there might be higher equilibrium metabolite concentrations)?

- At the end of p7, it reads 'If the reproducing taxon has more than one exchange (...), an equal amount of metabolites are made available to each recipient taxon.' I'm assuming this means that the available metabolites are equally divided among recipients, keeping the total amount the same? (in contrast to giving each recipient all the secreted metabolites?) Please clarify in the text.

- So the uptake by cross-feeding is not affected by competition coefficients, why not? This seems an important assumption. To what extent do the results hold when secondary metabolites are divided among cross-feeders proportional to their competition coefficients?

- There seems to be no hierarchy in the complexity of the produced metabolites. Using the example as is given in the introduction, does this imply that glucose metabolism resulting in the production of acetate as byproduct, is equally likely as acetate consumption resulting in the production of glucose, and that both processes can simultaneously take place in different taxa? How realistic is this? I would expect some kind of hierarchy, where less complex metabolites are byproducts from the consumption of more complex molecules.

- At what concentration is the invader introduced and when is an invasion considered to be successful (e.g. when it initially increases in abundance, or when it reaches a certain threshold)? These seem important details that I missed in the text.

- On p11: 'Secondary invaders (those introduced after the first invader had succeeded or failed) were more successful than primary invaders...'. I can see how this can happen if the first invader is successful, as this leads to an increase in the number of available metabolites. But how can this be the case when the first invader has failed? In that case, community and resource concentrations remain at their equilibrium, so why becomes a secondary invader, on average, more successful here?

- What exactly is the 'average redundancy of limiting flows'? (Fig. 3) How can these numbers be below 1? Again, here it would really help to mathematically show how this measure is obtained. Because I couldn't find in the text how this measure was obtained, I was unable to understand the (seemingly interesting) patterns shown in Fig. 3a.

- How can a successful invasion increase the number of coexisting taxa (Fig. 5a)? Taxa that have been excluded can somehow reappear after invasion? Does the 'change in total individuals' (Fig. 5b) include the invader? Again here, I couldn't find a quantitative description of what is meant by the 'change in metabolite flows' and the 'change in limiting flows'. This information should be given in the Methods section.

- Running the R-code results in the warning 'In lim.nut[position] <- paste(tx, which(reqs[, tx]) == ... : number of items to replace is not a multiple of replacement length'.

References:

Kettle, Helen, et al. "microPop: Modelling microbial populations and communities in R." *Methods*

in Ecology and Evolution 9.2 (2018): 399-409.

Marsland III, Robert, et al. "Available energy fluxes drive a transition in the diversity, stability, and functional structure of microbial communities." PLoS computational biology 15.2 (2019): e1006793.

Review form: Reviewer 2

Recommendation

Major revision is needed (please make suggestions in comments)

Scientific importance: Is the manuscript an original and important contribution to its field?

Excellent

General interest: Is the paper of sufficient general interest?

Excellent

Quality of the paper: Is the overall quality of the paper suitable?

Good

Is the length of the paper justified?

Yes

Should the paper be seen by a specialist statistical reviewer?

No

Do you have any concerns about statistical analyses in this paper? If so, please specify them explicitly in your report.

No

It is a condition of publication that authors make their supporting data, code and materials available - either as supplementary material or hosted in an external repository. Please rate, if applicable, the supporting data on the following criteria.

Is it accessible?

Yes

Is it clear?

Yes

Is it adequate?

Yes

Do you have any ethical concerns with this paper?

No

Comments to the Author

Herren investigates how the strength of cross-feeding and competition interactions between microbial taxa affects the susceptibility of a microbial community to invasion, and the consequences of invasion for community composition and structure. Using a model that explicitly simulates the dynamics of metabolites, the main results are that a) risk of invasion is greater when cross-feeding and competition for metabolites are weaker, and b) past invasions

increase the likelihood of future invasions as a result of the changes in the metabolite exchange networks of the native community following a successful primary invasion.

I enjoyed reading the manuscript. It is well-written, and the topic investigated is of great interest and relevance to microbial population biology and microbiome research. I do really appreciate the new insights provided by this study, but have two main concerns that I think need to be clarified.

Main comments

Basically, my main comments are about some of the key assumptions of the model, and their potential consequences for the results.

1) The model assumes that the invading taxa can compete for metabolites but have no cross-feeding relationships (as described in the Methods, page 9). I think that the lack of cross-feeding by invaders is a crucial assumption in the model. How would the results be affected if the invader can cross-feed? Could the findings that invaded communities have greater metabolites availability, less metabolites exchanged, and lower productivity, all be due to the lack of cross-feeding from the invader?

I think that this assumption needs to be made very clear throughout the text and discussed in more depth. I would also like to see some simulations showing how cross-feeding by the invader influences invasion outcome and community structure.

And for clarity, I would also add a schematic showing how invader vs native taxa differ in their metabolic profile (e.g. in figure 1a).

2) The author finds that a successful invasion generally decreases diversity. Given that communities are initially seeded with a fixed number of taxa (x at $t=0$) and assembled under no migration - i.e. no new taxa can enter the system except for the single invading taxon, then diversity will either remain the same (invader replaces another taxon), decrease, or increase but only by one taxon (that is, the invader). Thus, after a successful invasion, the number of taxa in the invaded community will never be greater than the number of taxa in the resident community at equilibrium +1. Is this interpretation accurate? If so, then the finding that invasion generally decreases diversity is not that surprising, and this should therefore be explicitly discussed.

Minor comments

3) Page 9. The statement “.. cross-feeding exchanges often need time to develop (e.g. time for proper spatial configuration [18], construction of nanotubes [19], or within-host coevolution [20]), and that an invading taxon would therefore have no preexisting cross-feeding relationships.” I think that this statement is not fully accurate. There is plenty of evidence that cross-feeding interactions between microbes can readily happen in nature without any pre-existing adaptation, as for instance, when microbes use the metabolic waste products of other microbes (i.e. ‘accidental cross-feeding’). So some statement mentioning that cross-feeding can also happen without adaptation would be more precise.

4) Page 10. The author states “Results were qualitatively similar regardless of the number of taxa used in the simulation, so long as there were sufficiently many taxa.” Is there any evidence supporting this statement?

5) Page 13, and figure 3. How is the ‘average redundancy’ calculated?

6) Page 13. The author states “increased susceptibility to invasion was greatest when less than one taxon provided each limiting resource”. How is having less than one taxon possible here?

Please clarify.

Decision letter (RSPB-2019-1494.R0)

13-Aug-2019

Dear Dr Herren:

I am writing to inform you that your manuscript RSPB-2019-1494 entitled "Disruption of cross-feeding interactions by invading taxa can cause invasional meltdown in microbial communities" has, in its current form, been rejected for publication in Proceedings B.

This action has been taken on the advice of referees, who have recommended that substantial and important revisions are necessary. With this in mind we would be happy to consider a resubmission, provided the comments of the referees are fully addressed. However please note that this is not a provisional acceptance.

Sincerely,
Professor Hans Heesterbeek
<mailto:proceedingsb@royalsociety.org>

Associate Editor
Board Member: 1
Comments to Author:

Thank you for your contribution to the special issue. Your manuscript has now been thoroughly read by myself and two reviewers. We all see great merit in the work, and feel it has clear potential to be a strong contribution to the literature. The work is timely, fills an important gap in our knowledge, and is likely to be well-received. That said, both reviewers have raised significant concerns that must be addressed before the work can be accepted. In particular, reviewer 1 felt that far more detail would be needed in terms of the modeling approach before the paper can be accurately assessed and more importantly that more detailed description of the model is required

for this manuscript to be useful to readers. This reviewer also has a number of excellent specific suggestions for improvement. Reviewer 2 has two additional concerns, the first relating to a key assumption of your model and the second relating to the ecological realism of one of the key findings. Overall, both reviewers saw promise in the model and liked the paper, but felt a major revision would be required before it could be published. I agree with the assessments and recommend that you take the needed time to revise appropriately. Some of the comments will be easy to address, but some require a more substantial revision, but I feel that once addressed, the suggestions will make for a much stronger paper that will be an excellent fit in the issue.

Reviewer(s)' Comments to Author:

Referee: 1

Comments to the Author(s)

The manuscript 'Disruption of cross-feeding interactions by invading taxa can cause invasional meltdown in microbial communities' uses a metabolite-explicit model to explore how the strength of cross-feeding in a microbial community affects the risk and consequences of invasions. This is an interesting topic, the manuscript is well-written and some interesting results are presented.

My main concern is that the manuscript lacks any mathematical description of the model and of model outcomes. Although an overview of all parameters is given (Fig. 1b) as well as some explanation in the main text (p. 6), without such a mathematical description (e.g. differential equations), fully understanding and reproducing the model, as well as comparing it to similar models (e.g. Kettle et al., 2018, Marsland et al., 2019), is very challenging. Especially as this is a purely theoretical paper, the theoretical framework should be presented in a clear and unambiguous way. I appreciate that R-code is accessible, but very few readers will go through almost 1000 lines of code to find relevant details. I also think that Fig. 1a, where a graphical overview of the model is given, can be improved, as it currently fails to visualize some important aspects of the model in a clear way. For instance, it is not clear from the figure what the metabolite requirements are for the different depicted taxa (why are there seemingly already some metabolites consumed before uptake takes place?), how taxa differ in competitive strength and how this affects their uptake, why some taxa grow while others do not, and what the cross-feeding interactions are. Further, I would suggest adding the symbols (as used in the text) describing the input parameters to Fig. 1b, and specifying (mathematically) how model output values are obtained (not all model outputs seem to be mentioned in Fig. 1b, e.g. redundancy of limiting flows?).

I do see the potential of this paper, and some interesting analysis and results are presented. However, I really need a better presentation of the modeling framework before I can start understanding and interpreting the results. In addition, I have the following specific comments/questions (with most of these reflecting the difficulties I had in understanding the model):

- I don't fully understand the cross-feeding procedure. A proportion p of all possible cross-feeding interactions is realized, but what exactly happens with the secreted metabolites that are not involved in a cross-feeding interaction? In the manuscript it says that these enter into the environment for competitive uptake. Can these directly be consumed by other taxa? If so, what is the effect of these cross-feeding interactions; is it giving priority to some taxa, before all other taxa get the chance of consuming the produced metabolites? But these taxa that consume the metabolites in this second phase surely are also 'cross-feeders', as they consume metabolites produced by others? Is this in agreement with what we observe in microbial communities: are there fixed cross-feeding interactions, even in the presence of more competitive taxa that are also capable of consuming the involved metabolites? (due to some space structure?)

- Related to this, how does the other half of the invasion landscapes (Fig. 2) look, if the proportion of cross-feeding increases further up to 1? Why is 0.5 chosen as the maximum? When set at 1, this would correspond to a scenario where all taxa can directly consume the secreted metabolites? Would this decrease the number of coexisting taxa (as only the competitive taxa can persist), and increase the invasion success (as the invader is even more competitive, and there might be higher equilibrium metabolite concentrations)?

- At the end of p7, it reads 'If the reproducing taxon has more than one exchange (...), an equal amount of metabolites are made available to each recipient taxon.' I'm assuming this means that the available metabolites are equally divided among recipients, keeping the total amount the same? (in contrast to giving each recipient all the secreted metabolites?) Please clarify in the text.

- So the uptake by cross-feeding is not affected by competition coefficients, why not? This seems an important assumption. To what extent do the results hold when secondary metabolites are divided among cross-feeders proportional to their competition coefficients?

- There seems to be no hierarchy in the complexity of the produced metabolites. Using the example as is given in the introduction, does this imply that glucose metabolism resulting in the production of acetate as byproduct, is equally likely as acetate consumption resulting in the production of glucose, and that both processes can simultaneously take place in different taxa? How realistic is this? I would expect some kind of hierarchy, where less complex metabolites are byproducts from the consumption of more complex molecules.

- At what concentration is the invader introduced and when is an invasion considered to be successful (e.g. when it initially increases in abundance, or when it reaches a certain threshold)? These seem important details that I missed in the text.

- On p11: 'Secondary invaders (those introduced after the first invader had succeeded or failed) were more successful than primary invaders...'. I can see how this can happen if the first invader is successful, as this leads to an increase in the number of available metabolites. But how can this be the case when the first invader has failed? In that case, community and resource concentrations remain at their equilibrium, so why becomes a secondary invader, on average, more successful here?

- What exactly is the 'average redundancy of limiting flows'? (Fig. 3) How can these numbers be below 1? Again, here it would really help to mathematically show how this measure is obtained. Because I couldn't find in the text how this measure was obtained, I was unable to understand the (seemingly interesting) patterns shown in Fig. 3a.

- How can a successful invasion increase the number of coexisting taxa (Fig. 5a)? Taxa that have been excluded can somehow reappear after invasion? Does the 'change in total individuals' (Fig. 5b) include the invader? Again here, I couldn't find a quantitative description of what is meant by the 'change in metabolite flows' and the 'change in limiting flows'. This information should be given in the Methods section.

- Running the R-code results in the warning 'In lim.nut[position] <- paste(tx, which(reqs[, tx]) == ... : number of items to replace is not a multiple of replacement length'.

References:

Kettle, Helen, et al. "microPop: Modelling microbial populations and communities in R." *Methods in Ecology and Evolution* 9.2 (2018): 399-409.

Marsland III, Robert, et al. "Available energy fluxes drive a transition in the diversity, stability, and functional structure of microbial communities." *PLoS computational biology* 15.2 (2019): e1006793.

Referee: 2

Comments to the Author(s)

Herren investigates how the strength of cross-feeding and competition interactions between microbial taxa affects the susceptibility of a microbial community to invasion, and the consequences of invasion for community composition and structure. Using a model that explicitly simulates the dynamics of metabolites, the main results are that a) risk of invasion is greater when cross-feeding and competition for metabolites are weaker, and b) past invasions increase the likelihood of future invasions as a result of the changes in the metabolite exchange networks of the native community following a successful primary invasion.

I enjoyed reading the manuscript. It is well-written, and the topic investigated is of great interest and relevance to microbial population biology and microbiome research. I do really appreciate the new insights provided by this study, but have two main concerns that I think need to be clarified.

Main comments

Basically, my main comments are about some of the key assumptions of the model, and their potential consequences for the results.

1) The model assumes that the invading taxa can compete for metabolites but have no cross-feeding relationships (as described in the Methods, page 9). I think that the lack of cross-feeding by invaders is a crucial assumption in the model. How would the results be affected if the invader can cross-feed? Could the findings that invaded communities have greater metabolites availability, less metabolites exchanged, and lower productivity, all be due to the lack of cross-feeding from the invader?

I think that this assumption needs to be made very clear throughout the text and discussed in more depth. I would also like to see some simulations showing how cross-feeding by the invader influences invasion outcome and community structure.

And for clarity, I would also add a schematic showing how invader vs native taxa differ in their metabolic profile (e.g. in figure 1a).

2) The author finds that a successful invasion generally decreases diversity. Given that communities are initially seeded with a fixed number of taxa (x at $t=0$) and assembled under no migration - i.e. no new taxa can enter the system except for the single invading taxon, then diversity will either remain the same (invader replaces another taxon), decrease, or increase but only by one taxon (that is, the invader). Thus, after a successful invasion, the number of taxa in the invaded community will never be greater than the number of taxa in the resident community at equilibrium +1. Is this interpretation accurate? If so, then the finding that invasion generally decreases diversity is not that surprising, and this should therefore be explicitly discussed.

Minor comments

3) Page 9. The statement “.. cross-feeding exchanges often need time to develop (e.g. time for proper spatial configuration [18], construction of nanotubes [19], or within-host coevolution [20]), and that an invading taxon would therefore have no preexisting cross-feeding relationships.” I think that this statement is not fully accurate. There is plenty of evidence that cross-feeding interactions between microbes can readily happen in nature without any pre-existing adaptation, as for instance, when microbes use the metabolic waste products of other microbes (i.e. ‘accidental cross-feeding’). So some statement mentioning that cross-feeding can also happen without adaptation would be more precise.

4) Page 10. The author states “Results were qualitatively similar regardless of the number of taxa

used in the simulation, so long as there were sufficiently many taxa. " Is there any evidence supporting this statement?

5) Page 13, and figure 3. How is the 'average redundancy' calculated?

6) Page 13. The author states "increased susceptibility to invasion was greatest when less than one taxon provided each limiting resource". How is having less than one taxon possible here? Please clarify.

Author's Response to Decision Letter for (RSPB-2019-1494.R0)

See Appendix A.

RSPB-2019-2945.R0

Review form: Reviewer 1

Recommendation

Major revision is needed (please make suggestions in comments)

Scientific importance: Is the manuscript an original and important contribution to its field?

Good

General interest: Is the paper of sufficient general interest?

Excellent

Quality of the paper: Is the overall quality of the paper suitable?

Acceptable

Is the length of the paper justified?

Yes

Should the paper be seen by a specialist statistical reviewer?

No

Do you have any concerns about statistical analyses in this paper? If so, please specify them explicitly in your report.

No

It is a condition of publication that authors make their supporting data, code and materials available - either as supplementary material or hosted in an external repository. Please rate, if applicable, the supporting data on the following criteria.

Is it accessible?

Yes

Is it clear?

Yes

Is it adequate?

Yes

Do you have any ethical concerns with this paper?

No

Comments to the Author

The revised manuscript 'Disruption of cross-feeding interactions by invading taxa can cause invasional meltdown in microbial communities' has substantially improved. The simulation procedure is much better explained, and I particularly appreciate the new Table 1, which is extremely helpful, and the newly added supplementary information, showing some robustness checks (note that references to the manuscript figures seem incorrect in the supplementary information, referring to the wrong figures).

However, I am disappointed that the author was not able to follow my suggestion for providing a mathematical description of the model (or at least did an attempt to capture some of the processes in equations), and I don't agree with the reasoning for why such a mathematical description would not be available in this case. It is totally possible to formulate a general mathematical model even if some parameters vary each run, by using general expressions (uptake of resource i by species j , excretion of resource i by species j , etc.). I also don't see why the property of microbes storing resources, would make it impossible to write the model in equations. It might add some complexity, but in principle, it surely should be possible to mathematically describe dynamics of both 'stored resources' and 'newly added' resources, together affecting microbial growth? Also, Fig. 1c-d highly resembles typical output from a set of coupled differential equations. I still strongly believe that adding a mathematical description would make a much stronger paper, for the reasons I listed in my previous review. Explaining the model only verbally also makes it more sensitive to misinterpretation. Indeed, this is reflected in several of the previous comments raised by me and Referee 2, showing how unprecise wording could lead to confusion and misunderstanding. This is less likely to happen if exact definitions are given.

To what extent do recently proposed Consumer-Resource models (e.g. Goldford et al., 2018, cited in the manuscript, and recent papers by Marsland), resemble the model proposed by Herren? Many of the relevant processes (e.g. influx of multiple resources, secondary metabolite excretion, and variation in microbial competitive abilities and resource preferences) are explicitly present in these equations. The most notable difference seems the distinction made by Herren between direct and indirect cross-feeding, but I would think that this could be implemented by splitting the resource uptake function into two components, one describing competitive uptake from the environment, and one describing uptake through fixed cross-feeding relationships. I have the impression that without much modification of already developed equations, it will be possible to fully capture the model proposed here. And even if I am mistaken in this, it would be extremely useful to (mathematically) show which aspects of the model here are different from earlier work. Again, especially because this is a purely theoretical study, I believe it is really a missed opportunity.

This having said, I do believe that the manuscript has greatly improved, and that it, also in its current form, provides some interesting, new insights on how levels of cross-feeding could affect susceptibility to invasions in microbial communities.

Review form: Reviewer 2**Recommendation**

Major revision is needed (please make suggestions in comments)

Scientific importance: Is the manuscript an original and important contribution to its field?
Good

General interest: Is the paper of sufficient general interest?
Good

Quality of the paper: Is the overall quality of the paper suitable?
Acceptable

Is the length of the paper justified?
Yes

Should the paper be seen by a specialist statistical reviewer?
No

Do you have any concerns about statistical analyses in this paper? If so, please specify them explicitly in your report.
No

It is a condition of publication that authors make their supporting data, code and materials available - either as supplementary material or hosted in an external repository. Please rate, if applicable, the supporting data on the following criteria.

Is it accessible?
Yes

Is it clear?
Yes

Is it adequate?
Yes

Do you have any ethical concerns with this paper?
No

Comments to the Author

The revisions made by Herren have significantly improved the clarity of the manuscript but I still have some concerns.

1) My major concern is how the term cross-feeding is defined in the paper. The model accounts for both 'direct' and 'indirect' cross feeding but only direct cross feeding is called crossfeeding while indirect crossfeeding is subsumed under the 'competition' category.

For instance, line 92 (Methods), the sentence "Taxa interact through competition for metabolites in the environment and through crossfeeding of metabolites" is potentially misleading because taxa do directly compete for metabolites in the environmental pool but those metabolites were actually produced by one taxa and consumed by another taxa, so they are in fact 'cross feeding' metabolites. Thus, saying that a taxa (native or invader) cannot crossed actually means that it cannot directly cross-feed but can indirectly crossfeed through the environmental pool.

Although a new sentence has been added to the revised manuscript to explain that crossfeeding in the paper only refers to direct cross feeding but that both direct and indirect cross-feeding are allowed in the model, it can easily be overlooked. This is such an important assumption (yet counterintuitive given that indirect crossfeeding is widespread in natural communities) that I think it should made very clear throughout the manuscript.

Also, this seems to suggest that stronger 'direct' cross feeding leads to less opportunities for indirect cross feeding, and thereby stronger competition. Does it mean that cross-feeding and competition are not independent of each other in the model? What is the role of indirect crossfeeding for the results? What if indirect cross-feeding is turned-off in the model?

- Related to this, line 111 it says "The cross-feeding step occurred separately from competitive uptake of metabolites from the environment". How realistic is this assumption?

2) Why assuming that native taxa that were previously absent can "reappear" after invasion? And why assuming a threshold of 1? How would the results change if the threshold was lower or higher?

Other comments:

3) To make the model more accessible (in light of the reviewer 1 comment), I would suggest adding a pseudo-code describing the steps/rules governing the model.

Decision letter (RSPB-2019-2945.R0)

07-Feb-2020

Dear Dr Herren:

Your manuscript has now been peer reviewed and the reviews have been assessed by an Associate Editor. The reviewers' comments (not including confidential comments to the Editor) and the comments from the Associate Editor are included at the end of this email for your reference. As you will see, the reviewers and the Associate Editor have raised some concerns with your manuscript and we would like to invite you to revise your manuscript to address them.

Research ethics:

Use of animals and field studies:

Please submit a copy of your revised paper within three weeks. If we do not hear from you within this time your manuscript will be rejected. If you are unable to meet this deadline please let us know as soon as possible, as we may be able to grant a short extension.

Best wishes,
Professor Hans Heesterbeek
mailto: proceedingsb@royalsociety.org

Associate Editor

Comments to Author:

Thank you for revising your manuscript for consideration in the special issue on microbiomes. The work has now been reviewed by myself and two reviewers, and we all feel that the revisions made have led to a significant improvement. That said, both reviewers have explained very clearly why they feel a mathematical description of the model, with clear and transparent information about the assumptions being made, is critical to the utility of the work to readers. There might be a compromise, where assumptions are more clearly laid out without a formal model presented, but I think the authors should seriously consider including a model as suggested by reviewer 1 if at all possible. In the end, both reviewers are highly positive about the work but they see the full potential as unmet, and offer further suggestions for how this could be done. I look forward to receiving a revised manuscript, and to including the work in the special issue.

Reviewer(s)' Comments to Author:

Referee: 1

Comments to the Author(s).

The revised manuscript 'Disruption of cross-feeding interactions by invading taxa can cause invasional meltdown in microbial communities' has substantially improved. The simulation procedure is much better explained, and I particularly appreciate the new Table 1, which is extremely helpful, and the newly added supplementary information, showing some robustness checks (note that references to the manuscript figures seem incorrect in the supplementary information, referring to the wrong figures).

However, I am disappointed that the author was not able to follow my suggestion for providing a mathematical description of the model (or at least did an attempt to capture some of the processes in equations), and I don't agree with the reasoning for why such a mathematical description would not be available in this case. It is totally possible to formulate a general mathematical model even if some parameters vary each run, by using general expressions (uptake of resource i by species j , excretion of resource i by species j , etc.). I also don't see why the property of microbes storing resources, would make it impossible to write the model in equations. It might add some complexity, but in principle, it surely should be possible to mathematically describe dynamics of both 'stored resources' and 'newly added' resources, together affecting microbial growth? Also, Fig. 1c-d highly resembles typical output from a set of coupled differential equations. I still strongly believe that adding a mathematical description would make a much stronger paper, for the reasons I listed in my previous review. Explaining the model only verbally also makes it more sensitive to misinterpretation. Indeed, this is reflected in several of the previous comments raised by me and Referee 2, showing how unprecise wording could lead to confusion and misunderstanding. This is less likely to happen if exact definitions are given.

To what extent do recently proposed Consumer-Resource models (e.g. Goldford et al., 2018, cited in the manuscript, and recent papers by Marsland), resemble the model proposed by Herren? Many of the relevant processes (e.g. influx of multiple resources, secondary metabolite excretion, and variation in microbial competitive abilities and resource preferences) are explicitly present in these equations. The most notable difference seems the distinction made by Herren between direct and indirect cross-feeding, but I would think that this could be implemented by splitting the resource uptake function into two components, one describing competitive uptake from the environment, and one describing uptake through fixed cross-feeding relationships. I have the impression that without much modification of already developed equations, it will be possible to

fully capture the model proposed here. And even if I am mistaken in this, it would be extremely useful to (mathematically) show which aspects of the model here are different from earlier work. Again, especially because this is a purely theoretical study, I believe it is really a missed opportunity.

This having said, I do believe that the manuscript has greatly improved, and that it, also in its current form, provides some interesting, new insights on how levels of cross-feeding could affect susceptibility to invasions in microbial communities.

Referee: 2

Comments to the Author(s).

The revisions made by Herren have significantly improved the clarity of the manuscript but I still have some concerns.

1) My major concern is how the term cross-feeding is defined in the paper. The model accounts for both 'direct' and 'indirect' cross feeding but only direct cross feeding is called crossfeeding while indirect crossfeeding is subsumed under the 'competition' category.

For instance, line 92 (Methods), the sentence "Taxa interact through competition for metabolites in the environment and through crossfeeding of metabolites" is potentially misleading because taxa do directly compete for metabolites in the environmental pool but those metabolites were actually produced by one taxa and consumed by another taxa, so they are in fact 'cross feeding' metabolites. Thus, saying that a taxa (native or invader) cannot cross actually means that it cannot directly cross-feed but can indirectly crossfeed through the environmental pool.

Although a new sentence has been added to the revised manuscript to explain that crossfeeding in the paper only refers to direct cross feeding but that both direct and indirect cross-feeding are allowed in the model, it can easily be overlooked. This is such an important assumption (yet counterintuitive given that indirect crossfeeding is widespread in natural communities) that I think it should be made very clear throughout the manuscript.

Also, this seems to suggest that stronger 'direct' cross feeding leads to less opportunities for indirect cross feeding, and thereby stronger competition. Does it mean that cross-feeding and competition are not independent of each other in the model? What is the role of indirect crossfeeding for the results? What if indirect cross-feeding is turned-off in the model?

- Related to this, line 111 it says "The cross-feeding step occurred separately from competitive uptake of metabolites from the environment". How realistic is this assumption?

2) Why assuming that native taxa that were previously absent can "reappear" after invasion? And why assuming a threshold of 1? How would the results change if the threshold was lower or higher?

Other comments:

3) To make the model more accessible (in light of the reviewer 1 comment), I would suggest adding a pseudo-code describing the steps/rules governing the model.

Author's Response to Decision Letter for (RSPB-2019-2945.R0)

See Appendix B.

RSPB-2019-2945.R1 (Revision)

Review form: Reviewer 1

Recommendation

Accept as is

Scientific importance: Is the manuscript an original and important contribution to its field?

Good

General interest: Is the paper of sufficient general interest?

Good

Quality of the paper: Is the overall quality of the paper suitable?

Good

Is the length of the paper justified?

Yes

Should the paper be seen by a specialist statistical reviewer?

No

Do you have any concerns about statistical analyses in this paper? If so, please specify them explicitly in your report.

No

It is a condition of publication that authors make their supporting data, code and materials available - either as supplementary material or hosted in an external repository. Please rate, if applicable, the supporting data on the following criteria.

Is it accessible?

Yes

Is it clear?

Yes

Is it adequate?

Yes

Do you have any ethical concerns with this paper?

No

Comments to the Author

I appreciate that the author included a general mathematical formulation of the model. I believe this manuscript would be a nice addition to the literature on microbial cross-feeding, and have no further comments.

Review form: Reviewer 2

Recommendation

Accept with minor revision (please list in comments)

Scientific importance: Is the manuscript an original and important contribution to its field?
Good

General interest: Is the paper of sufficient general interest?
Excellent

Quality of the paper: Is the overall quality of the paper suitable?
Good

Is the length of the paper justified?
Yes

Should the paper be seen by a specialist statistical reviewer?
No

Do you have any concerns about statistical analyses in this paper? If so, please specify them explicitly in your report.
No

It is a condition of publication that authors make their supporting data, code and materials available - either as supplementary material or hosted in an external repository. Please rate, if applicable, the supporting data on the following criteria.

Is it accessible?
Yes

Is it clear?
Yes

Is it adequate?
Yes

Do you have any ethical concerns with this paper?
No

Comments to the Author

The revised manuscript 'Disruption of cross-feeding interactions by invading taxa can cause invasional meltdown in microbial communities' has mostly addressed my previous comments. I have the following remaining points (most of these should be straightforward to address):

I appreciate the newly included mathematical formulation of the model. I am still a little disappointed that the author does little to explain (verbally or mathematically) how this model differs from previous studies (while I asked this explicitly in my previous comment). But I appreciate that all the mathematical details are now available for readers interested in making this comparison, or in extending this model. Please add a section heading, and consider including a table of contents to the SI.

L220. I am confused by what the author means by: '(meaning, with the same metabolic profile)'. If the second invader has the same metabolic profile as the primary invader in an invulnerable community, why can't the secondary invader always invade? ('an invulnerable community' means that it can be invaded by the primary invader, so why not by a secondary invader that has the same metabolic profile?) Please clarify.

Fig. 4. It is extremely hard to see which shading belongs to which color. There seems to be a lot of overlap, raising the question of how meaningful any differences in the median value are (at least

for some of the results). I would suggest to improve the readability of these graphs, maybe using colored lines instead of shaded envelopes? And perhaps only discuss those results that are significantly different, and/or mention the large overlap?

L292. "It was also possible...". But this doesn't happen here, correct? Perhaps move this sentence to the next section, where describing Fig. 5 (as there, the number of taxa coexisting indeed increases in some cases).

Lastly, I agree with Reviewer 2 that it should be made very clear that both direct and indirect cross-feeding occur in the model, but that the paper uses the term 'cross feeding' to refer to direct cross feeding only. In my opinion, the revised manuscript has made insufficient changes to address this important point (in fact, looking at the marked document, except for the addition to L92, there have been no additional edits to clarify this). I think it will be easy for readers to overlook this point (for instance, this distinction between direct and indirect cross-feeding is missing in the abstract). The questions raised by Reviewer 2 on the dependence between indirect and direct cross-feeding, and the effects of removing indirect cross-feeding, are interesting and relevant. I understand that it is unfeasible to run all these analysis at the high resolution used for the main figures, but is it possible to at least explore the direction of some of these effects (e.g. by using less replicates and/or using p increments of e.g. 0.05 instead of 0.01)?

Decision letter (RSPB-2019-2945.R1)

11-Apr-2020

Dear Dr Herren

I am pleased to inform you that your manuscript RSPB-2019-2945.R1 entitled "Disruption of cross-feeding interactions by invading taxa can cause invasional meltdown in microbial communities" has been accepted for publication in Proceedings B, pending some final minor revision.

The referees have recommended publication, but one reviewer also suggests some minor revisions to your manuscript. Therefore, I invite you to respond to the referee's comments and revise your manuscript. Because the schedule for publication is very tight, it is a condition of publication that you submit the revised version of your manuscript within 7 days. If you do not think you will be able to meet this date please let us know.

Sincerely,

Professor Hans Heesterbeek
 Editor, Proceedings B
 mailto:proceedingsb@royalsociety.org

Associate Editor:

Board Member: 1

Comments to Author:

Thank you for taking the time to review your manuscript so thoroughly in light of reviewer comments. As you will see, both reviewers think this will make a nice contribution to the literature, but reviewer 2 has a few minor comments that should be addressed prior to publication. I look forward to receiving your revision and including the work in the special issue.

Reviewer(s)' Comments to Author:

Referee: 2

Comments to the Author(s)

I appreciate that the author included a general mathematical formulation of the model. I believe this manuscript would be a nice addition to the literature on microbial cross-feeding, and have no further comments.

Referee: 1

Comments to the Author(s)

The revised manuscript 'Disruption of cross-feeding interactions by invading taxa can cause invasional meltdown in microbial communities' has mostly addressed my previous comments. I have the following remaining points (most of these should be straightforward to address):

I appreciate the newly included mathematical formulation of the model. I am still a little disappointed that the author does little to explain (verbally or mathematically) how this model differs from previous studies (while I asked this explicitly in my previous comment). But I appreciate that all the mathematical details are now available for readers interested in making this comparison, or in extending this model. Please add a section heading, and consider including a table of contents to the SI.

L220. I am confused by what the author means by: '(meaning, with the same metabolic profile)'. If the second invader has the same metabolic profile as the primary invader in an invulnerable community, why can't the secondary invader always invade? ('an invulnerable community' means that it can be invaded by the primary invader, so why not by a secondary invader that has the same metabolic profile?) Please clarify.

Fig. 4. It is extremely hard to see which shading belongs to which color. There seems to be a lot of overlap, raising the question of how meaningful any differences in the median value are (at least for some of the results). I would suggest to improve the readability of these graphs, maybe using colored lines instead of shaded envelopes? And perhaps only discuss those results that are significantly different, and/or mention the large overlap?

L292. "It was also possible...". But this doesn't happen here, correct? Perhaps move this sentence to the next section, where describing Fig. 5 (as there, the number of taxa coexisting indeed increases in some cases).

Lastly, I agree with Reviewer 2 that it should be made very clear that both direct and indirect cross-feeding occur in the model, but that the paper uses the term 'cross feeding' to refer to direct cross feeding only. In my opinion, the revised manuscript has made insufficient changes to address this important point (in fact, looking at the marked document, except for the addition to L92, there have been no additional edits to clarify this). I think it will be easy for readers to overlook this point (for instance, this distinction between direct and indirect cross-feeding is missing in the abstract). The questions raised by Reviewer 2 on the dependence between indirect and direct cross-feeding, and the effects of removing indirect cross-feeding, are interesting and relevant. I understand that it is unfeasible to run all these analysis at the high resolution used for the main figures, but is it possible to at least explore the direction of some of these effects (e.g. by using less replicates and/or using p increments of e.g. 0.05 instead of 0.01)?

Author's Response to Decision Letter for (RSPB-2019-2945.R1)

See Appendix C.

Decision letter (RSPB-2019-2945.R2)

16-Apr-2020

Dear Dr Herren

I am pleased to inform you that your manuscript entitled "Disruption of cross-feeding interactions by invading taxa can cause invasional meltdown in microbial communities" has been accepted for publication in Proceedings B.

Open Access

Paper charges

Sincerely,
Proceedings B
mailto: proceedingsb@royalsociety.org

Appendix A

Referee: 1

Comments to the Author(s)

The manuscript 'Disruption of cross-feeding interactions by invading taxa can cause invasional meltdown in microbial communities' uses a metabolite-explicit model to explore how the strength of cross-feeding in a microbial community affects the risk and consequences of invasions. This is an interesting topic, the manuscript is well-written and some interesting results are presented.

My main concern is that the manuscript lacks any mathematical description of the model and of model outcomes. Although an overview of all parameters is given (Fig. 1b) as well as some explanation in the main text (p. 6), without such a mathematical description (e.g. differential equations), fully understanding and reproducing the model, as well as comparing it to similar models (e.g. Kettle et al., 2018, Marsland et al., 2019), is very challenging. Especially as this is a purely theoretical paper, the theoretical framework should be presented in a clear and unambiguous way. I appreciate that R-code is accessible, but very few readers will go through almost 1000 lines of code to find relevant details. I also think that Fig. 1a, where a graphical overview of the model is given, can be improved, as it currently fails to visualize some important aspects of the model in a clear way. For instance, it is not clear from the figure what the metabolite requirements are for the different depicted taxa (why are there seemingly already some metabolites consumed before uptake takes place?), how taxa differ in competitive strength and how this affects their uptake, why some taxa grow while others do not, and what the cross-feeding interactions are. Further, I would suggest adding the symbols (as used in the text) describing the input parameters to Fig. 1b, and specifying (mathematically) how model output values are obtained (not all model outputs seem to be mentioned in Fig. 1b, e.g. redundancy of limiting flows?).

These comments all center around giving a more precise and understandable explanation of the model and its outputs. I believe a challenge in giving mathematical specifications in this case is that the model does not follow any mathematical formulation, due to 1) the random selection of taxa in each model run and 2) the fact that each cell can store 1 unit of each metabolite. These features make it tractable to build and analyze a model containing cross-feeding and invasion, but they come at the expense of any clean mathematical description. I would instead describe the model as imposing a set of "rules" for community assembly, and then running many simulated communities to see how these rules affect how communities play out.

Your point that the framework could be easier to follow is well taken. I have made several changes throughout the manuscript in accordance with your suggestions. I believe these changes, along with those in response to your specific comments, have substantially improved the readability of the manuscript. First, I revised Fig. 1a to have one panel giving the metabolite profiles and cross-feeding relationships present in the conceptual model. The conceptual model is now Fig. 1b:

a Model Specifications

Fig. 1: Design and output of simulation model studying invaders in microbial communities. Panel (a) gives model specifications for a simplified version of the cross-feeding model, containing three taxa, which is depicted in panel (b). Panel (b) shows the processes that occur during each timestep of the model. Different metabolites are represented by differently colored stars. Different taxa are represented by differently colored ovals. When a cell acquires one unit of each of its required metabolites, it reproduces and also excretes its given metabolites. In this example, the native community of three taxa has reached equilibrium. Panels (c) and (d) show results of model simulations, tracking both taxon abundances and the concentration of each metabolite in the environment through time. Panel (c) shows a successful invasion, where the invading taxon (pink line) persists in the community, whereas in (d), the invader is excluded from the community. Red lines indicate the time point when the invader is introduced.

Additionally, I have defined the output variables that are used throughout the results. The table of input and output variables (previously Fig. 1b) is now Table 1. Here, I now list the outcomes recorded in the model (used in figures and captions) and explain how they are measured:

Table 1: Input Parameters and Measured Outputs for Cross-feeding Model

Input Parameters	Value
Maximum number of taxa in community (x)	20
Number of possible metabolites (m)	8
Number of metabolites required by each taxon (n)	5
Number of metabolites excreted by each taxon (q)	3
Flushing rate of cells and metabolites (f)	0.1
Metabolite input rate (i)	200 per timestep for each metabolite
Proportion of direct cross-feeding relationships (p)	0.0 to 0.5 in increments of 0.01
Mean competition coefficient of native taxa (c)	0.5 to 0.8 in increments of 0.01
Standard deviation of competition coefficients (v)	0.3 * mean competition coefficient (c)
Measured Model Outputs	Definition
Persistence of invader	An invasion was deemed successful if the invader had an abundance greater than 1 at model equilibrium
Total individuals	Sum of all individuals from all taxa at equilibrium
Taxa coexisting	Number of taxa with at least 1 individual present at equilibrium
Metabolites at equilibrium	Sum of all metabolites present in the environment at equilibrium
Metabolites traded	Sum of all metabolites directly exchanged through cross-feeding
Flows per taxon	Number of direct cross-feeding relationships per taxon
Redundancy of limiting flows	Average number of cross-feeding relationships that provide the growth-limiting nutrient to each taxon

I also now explicitly state there are 5000 sets of model outcomes for each parameter set, coming from the 5000 simulated communities (L 193 – 194):

Thus, there are 5000 values of each model output for each set of parameters evaluated.

Additionally, in order to make it clear that the model puts into place a random set of metabolites and taxa in each run, I added the following text describing how the metabolite profiles were selected (L 93 – 103):

At the beginning of each model run, x native taxa were introduced into the community. For example, for the models presented here, there were 20 native taxa, each with an abundance of 50, at the start of the simulation. There were 8 total metabolites, and each taxon required 5 of those 8 metabolites for reproduction. Thus, there were a total of 8 choose 5 (equal to 56) distinct niches that taxa could

occupy. From these 56 niches, 20 niches were randomly assigned to the native taxa, and one was assigned to the invasive taxon; this yielded 56 choose 20 (upwards of 100 trillion) combinations of possible metabolite requirements for the native taxa. Each taxon also excretes a subset of q metabolites, which do not overlap with its n required metabolites. The “input” metabolites were a set of n metabolites that entered the environment at the beginning of each time step, and one of the x native taxa had metabolite requirements that matched the input metabolites.

I do see the potential of this paper, and some interesting analysis and results are presented. However, I really need a better presentation of the modeling framework before I can start understanding and interpreting the results. In addition, I have the following specific comments/questions (with most of these reflecting the difficulties I had in understanding the model):

- I don't fully understand the cross-feeding procedure. A proportion p of all possible cross-feeding interactions is realized, but what exactly happens with the secreted metabolites that are not involved in a cross-feeding interaction? In the manuscript it says that these enter into the environment for competitive uptake. Can these directly be consumed by other taxa? If so, what is the effect of these cross-feeding interactions; is it giving priority to some taxa, before all other taxa get the chance of consuming the produced metabolites? But these taxa that consume the metabolites in this second phase surely are also 'cross-feeders', as they consume metabolites produced by others? Is this in agreement with what we observe in microbial communities: are there fixed cross-feeding interactions, even in the presence of more competitive taxa that are also capable of consuming the involved metabolites? (due to some space structure?)

These comments helped me realize that I had not distinguished between “direct” cross-feeding, whereby taxa transfer metabolites preferentially to another taxon, and “indirect” cross-feeding, where taxa secrete metabolites into the environment that can be taken up by other organisms. I have added text to clarify this, and also added text to further describe how cross-feeding relationships are determined (L 104 – 109 and L 132 – 140):

Cross-feeding in the model was implemented as one taxon directly transferring its excreted metabolites to another taxon that required those metabolites. All possible unidirectional metabolite transfers were identified by looking at which metabolites were excreted and required by all taxa; a random fraction (given by the cross-feeding parameter p) of these possible metabolite transfers were implemented as cross-feeding relationships in the model. The cross-feeding step occurred separately from competitive uptake of metabolites from the environment.

If these individuals are from taxa participating in cross-feeding, the excreted metabolites are preferentially available to the recipient taxon; in this case, the metabolites are directly transferred to the recipient without being available for competitive uptake. Any excreted metabolites that are not part of cross-feeding

relationships enter the environmental pools of metabolites. Thus, this model also allows for “indirect” cross-feeding, wherein taxa can consume metabolites from the environment that were produced by a different taxon. However, the term “cross-feeding” in this paper refers to direct metabolite transfers between taxa.

- Related to this, how does the other half of the invasion landscapes (Fig. 2) look, if the proportion of cross-feeding increases further up to 1? Why is 0.5 chosen as the maximum? When set at 1, this would correspond to a scenario where all taxa can directly consume the secreted metabolites? Would this decrease the number of coexisting taxa (as only the competitive taxa can persist), and increase the invasion success (as the invader is even more competitive, and there might be higher equilibrium metabolite concentrations)?

There were a couple reasons why the maximum cross-feeding value was 0.5. Most importantly, the dynamics of the model largely were largely stable beyond this point. In other words, all native taxa were present, and the invader rarely succeeded. The lack of variation among these communities meant that they were not terribly interesting to analyze. Additionally, it seemed unlikely that the proportion of direct cross-feeding relationships in a community would approach 1 in any natural microbial community. This would mean that a single cell would provide metabolites directly to 30-40 other cells, which seemed implausible.

- At the end of p7, it reads ‘If the reproducing taxon has more than one exchange (...), an equal amount of metabolites are made available to each recipient taxon.’ I’m assuming this means that the available metabolites are equally divided among recipients, keeping the total amount the same? (in contrast to giving each recipient all the secreted metabolites?) Please clarify in the text.

Yes, I see how this text was unclear, and have rephrased (L 135 – 136):

If the reproducing taxon has more than one cross-feeder, the excreted metabolites are divided equally among the recipient taxa.

- So the uptake by cross-feeding is not affected by competition coefficients, why not? This seems an important assumption. To what extent do the results hold when secondary metabolites are divided among cross-feeders proportional to their competition coefficients?

I have added these requested simulations to the supplementary material, and found that there is minimal change to model outputs when metabolites are distributed in proportion to competition coefficients. In response to these comments and those of Reviewer 2, I have added a paragraph to discuss these sensitivity analyses (L 375 – 392):

Invasive taxa often differ from native taxa in their interactions with other organisms [28]. In many cases, these altered biotic interactions contribute to the success of the invader ([27, 30]). The assumption in this model that invaders cannot

cross-feed is the primary way in which the invasive taxa are differentiated from native taxa. Although the invaders' competition coefficients were relatively high, they were still within the range of values that could be assigned to native taxa. This lack of cross-feeding by the invader proved crucial to the phenomenon of invasional meltdown; when allowing the invader to have the same cross-feeding dynamics as the native taxa, there was no increased susceptibility to future invasion after a primary invasion (Figs. S1). Furthermore, a successful invasion under these circumstances was less disruptive to overall community structure (Figs. S2 and S3). Thus, these sensitivity analyses show that even a single taxon that does not participate in cross-feeding strongly affects the entire microbial community. However, the model was much less sensitive to assumptions about how cross-feeding was implemented among native taxa, as results were qualitatively similar when native taxa were allowed to be differentially good or poor at obtaining metabolites through cross-feeding (Figs. S4, S5, S6). Thus, the conclusions from this study apply primarily to cases where the invader is not well integrated into metabolite exchanges among the native community. Future models might use different criteria to differentiate an invader from a native taxon, such as specifying unique metabolite requirements for the invader.

- There seems to be no hierarchy in the complexity of the produced metabolites. Using the example as is given in the introduction, does this imply that glucose metabolism resulting in the production of acetate as byproduct, is equally likely as acetate consumption resulting in the production of glucose, and that both processes can simultaneously take place in different taxa? How realistic is this? I would expect some kind of hierarchy, where less complex metabolites are byproducts from the consumption of more complex molecules.

I agree that this type of hierarchy is likely found in nature, but I believe it is too complex to introduce such dynamics into this model. Doing so would require several more parameters governing metabolism, and the effects of these extra parameters would be difficult to explore because the simulations require so many runs. Additionally, the metabolites would no longer all be equal to one another, so it would be difficult to adjust metabolite profiles to account for the discrepancy between metabolites. I think that this is an area where the model is intentionally simplistic, in order to make it computationally tractable.

- At what concentration is the invader introduced and when is an invasion considered to be successful (e.g. when it initially increases in abundance, or when it reaches a certain threshold)? These seem important details that I missed in the text.

I have added this definition to Table 1 to indicate that an invasion was deemed successful when it had an abundance of greater than 1 at equilibrium. I have added the following text to specify initial abundances (L 113 – 115):

The initial abundance of all native taxa when initializing the model was 50, and this was also the abundance at which the invader was introduced.

- On p11: 'Secondary invaders (those introduced after the first invader had succeeded or failed) were more successful than primary invaders...'. I can see how this can happen if the first invader is successful, as this leads to an increase in the number of available metabolites. But how can this be the case when the first invader has failed? In that case, community and resource concentrations remain at their equilibrium, so why becomes a secondary invader, on average, more successful here?

This wording was ambiguous, and I have clarified to rephrase (L 210):

Secondary invaders (those introduced at model equilibrium after the first invader)

- What exactly is the 'average redundancy of limiting flows'? (Fig. 3) How can these numbers be below 1? Again, here it would really help to mathematically show how this measure is obtained. Because I couldn't find in the text how this measure was obtained, I was unable to understand the (seemingly interesting) patterns shown in Fig. 3a.

I apologize for being inconsistent with how I referred to outcome variables. In large part, some of this confusion originated from the fact that "redundancy of cross-feeding relationships providing the limiting nutrient" was too large to put in the figure labels. Thus, I referred to cross-feeding relationships as metabolite "flows," but this was not clear in the previous version of the manuscript. I have now stated this explicitly in the text, and have defined the six outcome variables in Table 1 (L 176 – 183):

The community-level outcomes recorded were persistence of the invaders, total individuals in the community, number of taxa present in the community, and the number of metabolites in the environment at equilibrium (Table 1). The network-level outcomes recorded were the number of metabolites traded during each time step, the average number of cross-feeding relationships (abbreviated in figures as "flows" of metabolites) for each taxon, and the average number of cross-feeding relationships (again, metabolite "flows") providing each taxon's growth-limiting nutrient (Table 1).

- How can a successful invasion increase the number of coexisting taxa (Fig. 5a)? Taxa that have been excluded can somehow reappear after invasion? Does the 'change in total individuals' (Fig. 5b) include the invader? Again here, I couldn't find a quantitative description of what is meant by the 'change in metabolite flows' and the 'change in limiting flows'. This information should be given in the Methods section.

I have clarified as to how an invader can increase diversity (L 288 – 291):

It was also possible for an invader to lead to increased diversity by excreting novel metabolites into the environment, thereby creating new niches for taxa to occupy. In

this case, native taxa that were previously counted as absent (having a population of less than 1) increase in abundance to join the community.

Additionally, I have specified that invaders are tallied in the response variables (L 179 – 180):

Successful invaders were counted in the total number of individuals and total number of taxa.

Finally, I have changes Fig. 1b into Table 1, as to give definitions for all the response variables.

- Running the R-code results in the warning 'In lim.nut[position] <- paste(tx, which(reqs[, tx] == ... : number of items to replace is not a multiple of replacement length'.

I appreciate that you ran the code to ensure that it works. That warning occurs in the uncommon case that two metabolites are equally limiting to a population's growth rate. When that happens, the first of the growth-limiting resources is deemed the limiting metabolite.

References:

Kettle, Helen, et al. "microPop: Modelling microbial populations and communities in R." *Methods in Ecology and Evolution* 9.2 (2018): 399-409.

Marsland III, Robert, et al. "Available energy fluxes drive a transition in the diversity, stability, and functional structure of microbial communities." *PLoS computational biology* 15.2 (2019): e1006793.

Referee: 2

Comments to the Author(s)

Herren investigates how the strength of cross-feeding and competition interactions between microbial taxa affects the susceptibility of a microbial community to invasion, and the consequences of invasion for community composition and structure. Using a model that explicitly simulates the dynamics of metabolites, the main results are that a) risk of invasion is greater when cross-feeding and competition for metabolites are weaker, and b) past invasions increase the likelihood of future invasions as a result of the changes in the metabolite exchange networks of the native community following a successful primary invasion.

I enjoyed reading the manuscript. It is well-written, and the topic investigated is of great interest and relevance to microbial population biology and microbiome research. I do really appreciate the new insights provided by this study, but have two main concerns that I think need to be clarified.

Main comments

Basically, my main comments are about some of the key assumptions of the model, and their potential consequences for the results.

1) The model assumes that the invading taxa can compete for metabolites but have no cross-feeding relationships (as described in the Methods, page 9). I think that the lack of cross-feeding by invaders is a crucial assumption in the model. How would the results be affected if the invader can cross-feed? Could the findings that invaded communities have greater metabolites availability, less metabolites exchanged, and lower productivity, all be due to the lack of cross-feeding from the invader?

I think that this assumption needs to be made very clear throughout the text and discussed in more depth. I would also like to see some simulations showing how cross-feeding by the invader influences invasion outcome and community structure.

And for clarity, I would also add a schematic showing how invader vs native taxa differ in their metabolic profile (e.g. in figure 1a).

Yes, you are correct that the lack of cross-feeding by the invader is a crucial aspect of this model. From your comments, I believe that I was not sufficiently clear about this assumption in the prior version of the manuscript. I have made this aspect of the model more explicit in the text, and have added discussion of why this decision was made. Given that the model was designed to explore how cross-feeding mediates various parts of community assembly and invasion, I thought that having an invader that also impacted cross-feeding would further develop this framework of how metabolite exchange affects community dynamics.

Additionally, I have followed your suggestion and run the same model simulations under the condition where the invader is able to cross-feed. These results now appear as supplementary material. I think that it is important to note that, if the invader is able to cross-feed, then there is effectively little distinction between an invader and a native community member. Although the invader has a relatively high competition value, a native taxon could easily be assigned a similarly high value by chance. The invasive taxa do not systematically differ from native taxa in their metabolite requirements, as all metabolite profiles are randomly drawn from the same distribution. This is clarified in the first paragraph of the methods (L 160 – 168 and L 375 – 392).

The lack of cross-feeding relationships is the primary way in which the invader differs from native taxa. There are multiple reasons why invasive taxa were not allowed to cross-feed in the model. First, I reasoned that cross-feeding relationships often need time to develop (e.g. time for proper spatial configuration [18], construction of nanotubes [19], or within-host coevolution [20]), and that an invading taxon would therefore have no preexisting methods of directly acquiring metabolites. Additionally, many studies of invasive taxa have concluded that invasive taxa differ from native taxa in their biotic interactions (as reviewed in [28]). The lack of cross-feeding relationships for invaders differentiates the biotic interactions of invaders from those of native taxa.

Invasive taxa often differ from native taxa in their interactions with other organisms [28]. In many cases, these altered biotic interactions contribute to the success of the invader ([27, 30]). The assumption in this model that invaders cannot cross-feed is the primary way in which the invasive taxa are differentiated from native taxa. Although the invaders' competition coefficients were relatively high, they were still within the range of values that could be assigned to native taxa. This lack of cross-feeding by the invader proved crucial to the phenomenon of invasional meltdown; when allowing the invader to have the same cross-feeding dynamics as the native taxa, there was no increased susceptibility to future invasion after a primary invasion (Figs. S1). Furthermore, a successful invasion under these circumstances was less disruptive to overall community structure (Figs. S2 and S3). Thus, these sensitivity analyses show that even a single taxon that does not participate in cross-feeding strongly affects the entire microbial community. However, the model was much less sensitive to assumptions about how cross-feeding was implemented among native taxa, as results were qualitatively similar when native taxa were allowed to be differentially good or poor at obtaining metabolites through cross-feeding (Figs. S4, S5, S6). Thus, the conclusions from this study apply primarily to cases where the invader is not well integrated into metabolite exchanges among the native community. Future models might use different criteria to differentiate an invader from a native taxon, such as specifying unique metabolite requirements for the invader.

2) The author finds that a successful invasion generally decreases diversity. Given

that communities are initially seeded with a fixed number of taxa (x at $t=0$) and assembled under no migration - i.e. no new taxa can enter the system except for the single invading taxon, then diversity will either remain the same (invader replaces another taxon), decrease, or increase but only by one taxon (that is, the invader). Thus, after a successful invasion, the number of taxa in the invaded community will never be greater than the number of taxa in the resident community at equilibrium +1. Is this interpretation accurate? If so, then the finding that invasion generally decreases diversity is not that surprising, and this should therefore be explicitly discussed.

This is an interesting point that I had previously not drawn much attention toward. It is indeed possible for a successful invader to lead to more taxa joining the community. If the invader excretes metabolites that would otherwise not be present in the community, or would be present at minimal levels, this can create new niches for native taxa. The native taxa that were deemed "absent" had abundances of less than 1, but were not driven completely to zero. Thus, these taxa could re-enter the community if their niche was made available. I have clarified this in the following text (L 319 – 322):

It was also possible for an invader to lead to increased diversity by excreting novel metabolites into the environment, thereby creating new niches for taxa to occupy. In this case, native taxa that were previously counted as absent (having a population of less than 1) increase in abundance to join the community.

Minor comments

3) Page 9. The statement “.. cross-feeding exchanges often need time to develop (e.g. time for proper spatial configuration [18], construction of nanotubes [19], or within-host coevolution [20]), and that an invading taxon would therefore have no preexisting cross-feeding relationships. “

I think that this statement is not fully accurate. There is plenty of evidence that cross-feeding interactions between microbes can readily happen in nature without any pre-existing adaptation, as for instance, when microbes use the metabolic waste products of other microbes (i.e. ‘accidental cross-feeding’). So some statement mentioning that cross-feeding can also happen without adaptation would be more precise.

Yes, this is true that “accidental” or “indirect” cross-feeding is possible in natural communities, and also within this model. I have clarified that indirect cross-feeding occurs when taxa acquire metabolites from the environment that were produced by other taxa (L 137 – 140):

Any excreted metabolites that are not part of cross-feeding relationships enter the environmental pools of metabolites. Thus, this model also allows for “indirect” cross-feeding, wherein taxa can consume metabolites from the environment that were produced by a different taxon. However, the term “cross-feeding” in this paper refers to direct metabolite transfers between taxa.

4) Page 10. The author states “Results were qualitatively similar regardless of the number of taxa used in the simulation, so long as there were sufficiently many taxa.” Is there any evidence supporting this statement?

Before running the simulations on the computing cluster, I ran smaller groups of simulations to find a representative set of parameters for the models runs. During this testing period, I generally ran simulations with 10-30 taxa, and consistently recovered the same qualitative patterns as presented in the manuscript. However, because the simulations took a week or more to run at full scale, I could not systematically identify a cutoff for diversity where results started shifting due to the idiosyncrasies of communities assembled with only a few taxa.

5) Page 13, and figure 3. How is the ‘average redundancy’ calculated?

I’ve now included Table 1, where inputs and outputs of the model are defined. To answer your question here, the average redundancy is the average number of cross-feeding relationships that provide a taxon with its growth-limiting metabolite.

6) Page 13. The author states “increased susceptibility to invasion was greatest when less than one taxon provided each limiting resource”. How is having less than one taxon possible here? Please clarify.

Yes, I understand how this statement was unclear. I have rephrased to instead say (L 244 – 245):

I found that increased susceptibility to invasion was most common when the average number of taxa providing each limiting resource was less than 1 (Fig. 4a).

Appendix B

Associate Editor

Comments to Author:

Thank you for revising your manuscript for consideration in the special issue on microbiomes. The work has now been reviewed by myself and two reviewers, and we all feel that the revisions made have led to a significant improvement. That said, both reviewers have explained very clearly why they feel a mathematical description of the model, with clear and transparent information about the assumptions being made, is critical to the utility of the work to readers. There might be a compromise, where assumptions are more clearly laid out without a formal model presented, but I think the authors should seriously consider including a model as suggested by reviewer 1 if at all possible. In the end, both reviewers are highly positive about the work but they see the full potential as unmet, and offer further suggestions for how this could be done. I look forward to receiving a revised manuscript, and to including the work in the special issue.

I appreciate the effort that you and the reviewers have put in to help me improve this manuscript. I have now included a general mathematical model in the supplementary material to clarify the structure of the cross-feeding model. I did not believe it would fit within the main text, as the manuscript is already pushing the length limits, and the mathematical model is quite lengthy. I have responded to each reviewer criticism below, and thank the reviewers for their time in reading the revised manuscript.

Referee: 1

Comments to the Author(s).

The revised manuscript 'Disruption of cross-feeding interactions by invading taxa can cause invasional meltdown in microbial communities' has substantially improved. The simulation procedure is much better explained, and I particularly appreciate the new Table 1, which is extremely helpful, and the newly added supplementary information, showing some robustness checks (note that references to the manuscript figures seem incorrect in the supplementary information, referring to the wrong figures).

Apologies for the misnumbered references in the supplemental materials. I had originally inserted another figure during the revisions, leading to all figure numbers being off by 1. This is now corrected.

However, I am disappointed that the author was not able to follow my suggestion for providing a mathematical description of the model (or at least did an attempt to capture some of the processes in equations), and I don't agree with the reasoning for why such a mathematical description would not be available in this case. It is totally possible to formulate a general mathematical model even if some parameters vary each run, by using general expressions (uptake of resource i by species j , excretion of resource i by species j , etc.). I also don't see why the property of microbes storing resources, would make it impossible to write the model in equations. It might add some complexity, but in principle, it surely should be possible to mathematically describe dynamics of both 'stored resources' and 'newly added' resources, together affecting microbial growth? Also, Fig. 1c-d highly resembles typical output from a set of coupled differential equations. I still strongly believe that adding a mathematical description

would make a much stronger paper, for the reasons I listed in my previous review. Explaining the model only verbally also makes it more sensitive to misinterpretation. Indeed, this is reflected in several of the previous comments raised by me and Referee 2, showing how unprecise wording could lead to confusion and misunderstanding. This is less likely to happen if exact definitions are given.

To what extent do recently proposed Consumer-Resource models (e.g. Goldford et al., 2018, cited in the manuscript, and recent papers by Marsland), resemble the model proposed by Herren? Many of the relevant processes (e.g. influx of multiple resources, secondary metabolite excretion, and variation in microbial competitive abilities and resource preferences) are explicitly present in these equations. The most notable difference seems the distinction made by Herren between direct and indirect cross-feeding, but I would think that this could be implemented by splitting the resource uptake function into two components, one describing competitive uptake from the environment, and one describing uptake through fixed cross-feeding relationships. I have the impression that without much modification of already developed equations, it will be possible to fully capture the model proposed here. And even if I am mistaken in this, it would be extremely useful to (mathematically) show which aspects of the model here are different from earlier work. Again, especially because this is a purely theoretical study, I believe it is really a missed opportunity.

After reading these more detailed comments about what kind of representation was desired, I was able to formulate a general mathematical model for the assembly of a native community with cross-feeding. As it is quite lengthy, I have included it in the supplemental materials, and have stated at the beginning of the methods in the main text that a mathematical formulation is available (L 90-91):

A general mathematical formulation of the model is available in the supplementary materials, and is described here.

This having said, I do believe that the manuscript has greatly improved, and that it, also in its current form, provides some interesting, new insights on how levels of cross-feeding could affect susceptibility to invasions in microbial communities.

Referee: 2

Comments to the Author(s).

The revisions made by Herren have significantly improved the clarity of the manuscript but I still have some concerns.

1) My major concern is how the term cross-feeding is defined in the paper. The model accounts for both 'direct' and 'indirect' cross feeding but only direct cross feeding is called crossfeeding while indirect crossfeeding is subsumed under the 'competition' category.

For instance, line 92 (Methods), the sentence "Taxa interact through competition for metabolites in the environment and through crossfeeding of metabolites" is potentially misleading because taxa do directly compete for metabolites in the environmental pool but those metabolites were

actually produced by one taxa and consumed by another taxa, so they are in fact 'cross feeding' metabolites. Thus, saying that a taxa (native or invader) cannot cross feed actually means that it cannot directly cross-feed but can indirectly crossfeed through the environmental pool.

I have revised this line to give a definition of how cross-feeding is used in this manuscript (L92-93):

Taxa interact through competition for metabolites in the environment and through cross-feeding, defined here as the directed transfer of metabolites between taxa.

Although a new sentence has been added to the revised manuscript to explain that crossfeeding in the paper only refers to direct cross feeding but that both direct and indirect cross-feeding are allowed in the model, it can easily be overlooked. This is such an important assumption (yet counterintuitive given that indirect crossfeeding is widespread in natural communities) that I think it should be made very clear throughout the manuscript.

Also, this seems to suggest that stronger 'direct' cross feeding leads to less opportunities for indirect cross feeding, and thereby stronger competition. Does it mean that cross-feeding and competition are not independent of each other in the model? What is the role of indirect crossfeeding for the results? What if indirect cross-feeding is turned-off in the model?

Unfortunately, giving robust answers to these questions would require many weeks of simulation time, and thus these questions are outside the scope of the current manuscript. Briefly, if indirect cross-feeding were removed, the carrying capacity of the community would be strongly decreased, as fewer metabolites would be in circulation.

- Related to this, line 111 it says "The cross-feeding step occurred separately from competitive uptake of metabolites from the environment". How realistic is this assumption?

This model was inspired by the empirical observation that microbes can directly transfer metabolites between cells (Pande et al. 2015, Shitut et al. 2019), thereby making it impossible for other coexisting taxa to access those metabolites. Thus, at least in some circumstances, this type of direct cross-feeding is relevant to microbial community dynamics.

2) Why assuming that native taxa that were previously absent can "reappear" after invasion? And why assuming a threshold of 1? How would the results change if the threshold was lower or higher?

As mentioned in the previous round of review, removing the ability for taxa to re-colonize means that an invader will always decrease diversity. Allowing for taxa to be reintroduced made it possible to evaluate whether an invader could create additional niches for native taxa.

As for the question about the effects of implementing a threshold for presence or absence, I took this into account when determining the rate of input metabolites. The input rate was sufficiently high that virtually no taxa equilibrated at a value of less than 1. I thought that 1 was a reasonable threshold, since a single bacterial cell can reproduce.

Other comments:

3) To make the model more accessible (in light of the reviewer 1 comment), I would suggest adding a pseudo-code describing the steps/rules governing the model.

This was done and added to the supplemental materials.

Appendix C

Comments to the Author(s)

The revised manuscript 'Disruption of cross-feeding interactions by invading taxa can cause invasional meltdown in microbial communities' has mostly addressed my previous comments. I have the following remaining points (most of these should be straightforward to address):

I appreciate the newly included mathematical formulation of the model. I am still a little disappointed that the author does little to explain (verbally or mathematically) how this model differs from previous studies (while I asked this explicitly in my previous comment). But I appreciate that all the mathematical details are now available for readers interested in making this comparison, or in extending this model. Please add a section heading, and consider including a table of contents to the SI.

I have expanded the initial description within the supplemental materials to include page numbers where each subsection can be found.

L220. I am confused by what the author means by: '(meaning, with the same metabolic profile)'. If the second invader has the same metabolic profile as the primary invader in an invulnerable community, why can't the secondary invader always invade? ('an invulnerable community' means that it can be invaded by the primary invader, so why not by a secondary invader that has the same metabolic profile?) Please clarify.

I have added the following text to L 224-226:

Additionally, this analysis demonstrates that the metabolite profile of the invader is a determinant of invasion success, as communities that are invulnerable by one invader are not completely susceptible to a different invader.

Fig. 4. It is extremely hard to see which shading belongs to which color. There seems to be a lot of overlap, raising the question of how meaningful any differences in the median value are (at least for some of the results). I would suggest to improve the readability of these graphs, maybe using colored lines instead of shaded envelopes? And perhaps only discuss those results that are significantly different, and/or mention the large overlap?

I have changed the color scheme to enhance the contrast of the overlapping regions. When trying the overlaid lines approach, the lines could lie atop one another, making it even more difficult to ascertain where the boundaries lay. Additionally, because these results come from models with an arbitrary number of runs, any difference could be made statistically significant by increasing the number of runs. Thus, I think it is more in line with the modeling approach to discuss the relative magnitude of the differences across the various outcome variables.

L292. "It was also possible...". But this doesn't happen here, correct? Perhaps move this sentence to the next section, where describing Fig. 5 (as there, the number of

taxa coexisting indeed increases in some cases).

I have moved these two sentences to the discussion of Fig. 5, where this pattern is more obvious. They now appear on L 321 – 325.

Lastly, I agree with Reviewer 2 that it should be made very clear that both direct and indirect cross-feeding occur in the model, but that the paper uses the term ‘cross feeding’ to refer to direct cross feeding only. In my opinion, the revised manuscript has made insufficient changes to address this important point (in fact, looking at the marked document, except for the addition to L92, there have been no additional edits to clarify this). I think it will be easy for readers to overlook this point (for instance, this distinction between direct and indirect cross-feeding is missing in the abstract). The questions raised by Reviewer 2 on the dependence between indirect and direct cross-feeding, and the effects of removing indirect cross-feeding, are interesting and relevant. I understand that it is unfeasible to run all these analysis at the high resolution used for the main figures, but is it possible to at least explore the direction of some of these effects (e.g. by using less replicates and/or using p increments of e.g. 0.05 instead of 0.01)?

This point had been first addressed after the initial round of review, where I added additional text differentiating direct versus indirect cross-feeding (L 140 – 143). After the second round of review, I also added the change on L 92. I have added another sentence to the discussion to indicate that differences between indirect cross-feeding in native versus invasive taxa might be addressed in future work (L 394 – 397):

Future models might use different criteria to differentiate an invader from a native taxon, such as specifying unique metabolite requirements for the invader, or introducing distinctions between native and invasive taxa in their indirect cross-feeding.

Additionally, due to the tight schedule for revisions, there is insufficient time to conduct further simulations, and these questions are also outside the scope of the present study.